# Likelihood Matching for Diffusion Models

**Lei Qian** [1]  **Wu Su** [1]  **Yanqi Huang** [2]  **Song Xi Chen** [3]

## Abstract

We propose a Likelihood Matching approach for training diffusion models by first establishing an equivalence between the likelihood of the target data distribution and a likelihood along the sample path of the reverse diffusion. To efficiently compute the reverse sample likelihood, a quasi-likelihood is considered to approximate each reverse transition density by a Gaussian distribution with matched conditional mean and covariance, respectively. The score and Hessian functions for the diffusion generation are estimated by maximizing the quasi-likelihood, ensuring a consistent matching of both the first two transitional moments between every two time points. A stochastic sampler is introduced to facilitate computation that leverages both the estimated score and Hessian information. We establish consistency of the quasi-maximum likelihood estimation, and provide non-asymptotic convergence guarantees for the proposed sampler, quantifying the rates of the approximation errors due to the score and Hessian estimation, dimensionality, and the number of diffusion steps. Empirical and simulation evaluations demonstrate the effectiveness of the proposed Likelihood Matching and validate the theoretical results.

## 1. Introduction

Generative models and methods facilitate powerful learning of data distributions by generating controlled sequences of synthetic data, and stand as a cornerstone of modern machine learning, driving progress in areas like image synthesis, protein design, and data augmentation (Goodfellow et al., 2014; Sohl-Dickstein et al., 2015; Kobyzev et al., 2020; Watson et al., 2023; Dhariwal & Nichol, 2021; Yang et al., 2023; Chen et al., 2024). The mainstream diffusion methods like the denoising diffusion probabilistic models (DDPMs) (Ho et al., 2020) and the denoising diffusion implicit models (DDIMs) (Song et al., 2021a) have demonstrated state-of-the-art performance in generating high-fidelity samples, particularly in image synthesis (Betker et al., 2023; Esser et al., 2024). Among the leading methods, the score-based generative models (SGMs) (Sohl-Dickstein et al., 2015; Ho et al., 2020; Song et al., 2021c) have achieved remarkable success, producing synthetic samples across various domains. The models typically operate by progressively adding noise to data (forward process) and then learning to reverse this process (reverse process), often guided by estimating the score function (gradient of the log-likelihood) of the perturbed data distributions.

The standard training objective for SGMs is based on score matching (Hyvärinen, 2005; Vincent, 2011; Song et al., 2021c), which minimizes the discrepancy between a parameterized score function and the underlying score functions at different noise levels of the diffusion process. While being highly effective empirically, the score matching method provides only an indirect connection to the likelihood of the original data distribution $q_0$ as an upper bound rather than the likelihood itself. Maximizing the data likelihood directly is the approach for parameter estimation in Statistics, underpinned by attractive properties of the Maximum Likelihood Estimation (MLE), which often yields the most accurate estimators with desirable asymptotic properties like consistency and efficiency.

This paper explores a direct maximum likelihood framework for training diffusion models. We leverage a fundamental property that the path likelihood of the reverse diffusion process is intrinsically equivalent to the likelihood of the original data distribution $q_0(\theta)$ (up to constants related to the forward process) (Anderson, 1982; Haussmann & Pardoux, 1986) where $\theta$ denotes a parameter vector in a family of distributions $\mathcal{F}$ that $q_0$ belongs to. The equivalence (formalized in Proposition 3.1) suggests that maximizing the exact path likelihood of the reverse process is equivalent to maximizing the likelihood $\log q_0(\cdot; \theta)$.

[1] Center for Data Science, Peking University, Beijing, China [2] Guanghua School of Management, Peking University, Beijing, China [3] Department of Statistics and Data Science, Tsinghua University, Beijing, China. Correspondence to: Song Xi Chen <sxchen@tsinghua.edu.cn>.

*Proceedings of the 43rd International Conference on Machine Learning*, Seoul, South Korea. PMLR 306, 2026. Copyright 2026 by the author(s).

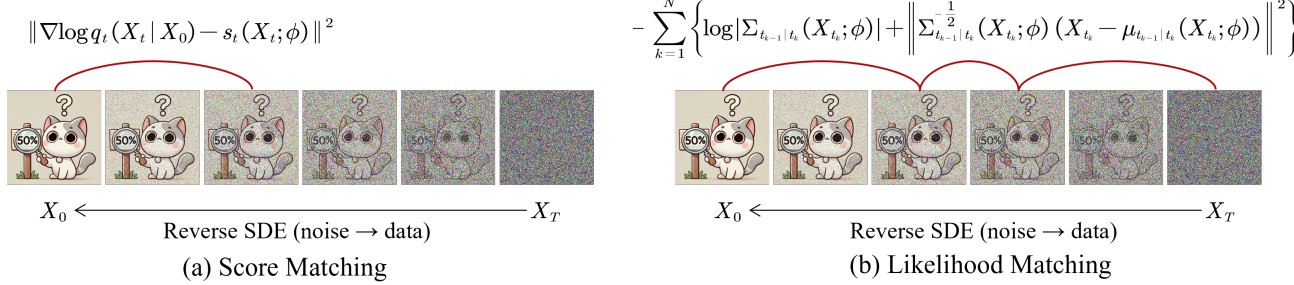

*Figure 1.* Illustration of Score Matching (a) versus Likelihood Matching (b) methods. The proposed Likelihood Matching captures a richer set of transition densities while incorporating both score matching and covariance matching, whereas Score Matching exclusively focuses on a single transition density and utilizes only first-order moment information.

To make it operational, we propose approximating the intractable reverse transition densities $p_{t-1|t}(Y_{t-1}|Y_t;\theta)$ via the Quasi-Maximum Likelihood Estimation (QMLE) with a proper Gaussian distributions (Wedderburn, 1974). As derived in Proposition 4.1, the mean and covariance of these conditional distributions depend not only on the score function $\nabla \log q_t(\cdot;\theta)$ but also on its Hessian function $\nabla^2 \log q_t(\cdot;\theta)$. We therefore parameterize both the score $s_t(\cdot;\phi)$ and the Hessian $H_t(\cdot;\phi)$ (e.g., using the neural networks) and optimize the parameters $\phi$ by minimizing the resulting approximate negative quasi-likelihood using the observed data trajectories.

Building upon this quasi-likelihood formulation, we introduce a computationally efficient objective called Likelihood Matching (LM). This objective not only provides a practical way to implement our framework but also offers a novel extension of the traditional score matching (SM), inherently incorporating covariance matching which is a form of likelihood weighting often beneficial in practice. Our key idea is summarized in Figure 1.

The main contributions of this work are the following

- We propose a novel training objective function for diffusion models based on the quasi-likelihood, leading to an approximation of the reverse path log-likelihood and a computationally efficient variant called Likelihood Matching (LM) that combines score matching and covariance matching with implicit likelihood weighting.

- We derive a stochastic sampler that leverages both the learned score function and Hessian information through the implied conditional mean and covariance structure of transition densities.

- We provide non-asymptotic convergence guarantees for the proposed sampler in total variation distance, characterizing the errors in terms of score and Hessian estimation error, dimension $d$, and diffusion steps $T$. It reveals that the reverse step error scales at $O(d^3 \log^{4.5} T/T)$, while the score estimation error and

Hessian estimation error are at the rate of $O(\sqrt{\log T})$ and $O(\log T/\sqrt{T})$, respectively.

- We theoretically demonstrate the consistency of the proposed quasi-maximum likelihood diffusion training under reverse quasi-likelihood objectives.

- We evaluate the proposed approach on standard benchmark image data, demonstrating its effectiveness and the impact of choices for Hessian approximation rank $r$ and the number of distinct transition probability densities evaluated per sample path, confirming the critical role of the learned Hessian through ablation studies.

Detailed proofs of theoretical results are provided in Appendix B.

## 2. Related Works

**Likelihood and Higher-Order Diffusion Objectives.** Recent works have incorporated likelihood information or higher-order terms into diffusion training, either via maximum-likelihood formulations for score-based models and diffusion ODEs (Song et al., 2021b; Lu et al., 2022; Zheng et al., 2023) or via Hessian-enhanced objectives (Dockhorn et al., 2022; Karras et al., 2022; Rissanen et al., 2024; Wang et al., 2025). However, these methods typically operate within regularized score-matching or probability-flow ODE formulations and thus optimize surrogate objectives or upper bounds rather than the data likelihood itself. By contrast, we start from the exact path likelihood of the reverse diffusion SDE and construct an analytical quasi-maximum likelihood approximation of the reverse transition densities, yielding an LM objective that directly targets data likelihood.

**Learning Reverse Covariance in Discrete-Time Diffusion.** Complementary to continuous-time approaches, a closely related line of work in discrete-time DDPMs focuses on learning or designing the reverse covariance via variational objectives. Representative examples include variance

interpolation (Nichol & Dhariwal, 2021), analytical ELBO-based derivations such as Analytic-DPM, SN-DDPM, and OCM-DDPM (Bao et al., 2022b; Ou et al., 2025; Bao et al., 2022a), and Gaussian-mixture refinements (Guo et al., 2023). While these methods improve likelihood and sampling by better parameterizing the covariance within a fixed ELBO framework, LM is formulated in the continuous-time reverse SDE setting and uses QML to approximate the full path likelihood, providing a distinct likelihood-based training paradigm rather than a covariance-tuning strategy.

## 3. Background and Motivations

### 3.1. Notations

Throughout the paper, we employ the following convention on notation: $\|\cdot\|$ designates the $L_2$ (spectral) norm for matrices or the $L_2$ norm for vectors, while $\|\cdot\|_F$ represents the Frobenius norm of a matrix. The determinant of a matrix is denoted by $|\cdot|$ or $\det(\cdot)$. For matrices $A$ and $B$, we use $\mathrm{tr}(A)$ to represent the trace of $A$, and $A \succeq B$ indicates that $A - B$ is positive semidefinite. For two probability measure $P$ and $Q$, we define their total-variation (TV) distance as $\mathrm{TV}(P\|Q) := \sup_{A \in \mathcal{F}} |P(A) - Q(A)|$ and their Kullback-Leibler (KL) divergence as $\mathrm{KL}(P\|Q) := \int \log(\mathrm{d}P/\mathrm{d}Q)\mathrm{d}P$. For two random vectors $X$ and $Y$, $X \overset{d}{=} Y$ signifies that their cumulative distribution functions $F_X$ and $F_Y$ are identical almost surely. We employ $X_{0:t}$ to represent the sequence $(X_0, X_1, \cdots, X_t)$, $q_{s|t}$ denotes the conditional probability density function (PDF) of $X_s$ given $X_t$ and $q_{0:t}$ represents the joint PDF of $X_{0:t}$. We use $f(x) \lesssim g(x)$ or $f(x) = O(g(x))$ (resp. $f(x) \gtrsim g(x)$) to denote $f(x) \leq cg(x)$ (resp. $f(x) \geq cg(x)$) for a universal constant $c$ and all $x$. We write $f(x) \asymp g(x)$ when both $f(x) \lesssim g(x)$ and $f(x) \gtrsim g(x)$ hold.

### 3.2. Preliminaries and Motivations

We adhere to the foundational generative models introduced in Song et al. (2021c), where both the forward and reverse processes are characterized by a unified system of stochastic differential equations (SDEs).

Let $X_1, \cdots, X_n$ be independent and identically distributed (IID) random observations from a target distribution on $\mathbb{R}^d$ with density $q_0$. We assume this distribution belongs to a specific parametric family of distributions $\mathcal{F}_\theta$, characterized by a true parameter $\theta \in \mathbb{R}^h$, with the PDF $q_0(\theta)$. The high dimensionality of $\theta$ often presents challenges for traditional statistical inference methods, highlighting a key area where diffusion models can provide better solutions.

The forward diffusion process for $\{X_t\}_{t \in [0,T]}$ in $\mathbb{R}^d$ is ex-

pressed by SDEs

$$\mathrm{d}X_t = -\frac{1}{2}\beta_t X_t \mathrm{d}t + \sqrt{\beta_t}\mathrm{d}W_t, \tag{1}$$

where $X_0 \sim q_0(\theta)$, $\beta_t$ is a given time-dependent diffusion coefficient and $W_t$ denotes the Brownian motion. Let $q_t(\cdot; \theta)$ represents the PDF of $X_t$. It is noted that $q_t(\cdot; \theta)$ only depend on $\theta$ since the transition density $q_{t|t-1}$ is free of the parameter $\theta$ as $\beta_t$ is known.

Under mild regularity conditions on $q_0(\theta)$, Anderson (1982) and Haussmann & Pardoux (1986) establish that there are reverse-time SDEs $\{Y_t\}_{t \in [T,0]}$ which exhibit identical marginal distributions as the forward diffusion processes (1) such that $Y_t \overset{d}{=} X_t$, and satisfy

$$\mathrm{d}Y_t = \frac{1}{2}\beta_t(Y_t + 2\nabla \log q_t(Y_t; \theta))\mathrm{d}t + \sqrt{\beta_t}\mathrm{d}\bar{W}_t, \tag{2}$$

where $Y_T \sim q_T(\theta)$, $\bar{W}_t$ is the Brownian motion, $p_t(\cdot; \theta)$ is the PDF of $Y_t$ and $\nabla \log q_t(\cdot; \theta)$ represents the score function of the marginal density $q_t$. As both $\theta$ and $q_t$ are unknown, the exact score function is inaccessible. Therefore, we endeavor to approximate it with a suitable estimator $s_t(\cdot)$. Typically, we parametrize $s_t(\cdot)$ as $s_t(\cdot; \phi)$ via either a neural network (or parametric models like the Gaussian Mixtures) base on the sample $\{X_i\}_{i=1}^n$. To be precise, throughout this paper, $\theta$ refers to the true parameters of the data distribution in an oracle setting (i.e., when the parametric family $\mathcal{F}_\theta$ is known), whereas $\phi$ denotes the learnable parameters of our neural network models.

Additionally, we substitute the distribution of $Y_T$ with a prior distribution $\pi$, which is specifically chosen as $\mathcal{N}_d(0, I_d)$ to facilitate data generation. Consequently, the modified reverse diffusion process $\{\hat{Y}_t\}_{t \in [T,0]}$ is defined as

$$\mathrm{d}\hat{Y}_t = \frac{1}{2}\beta_t(\hat{Y}_t + 2s_t(\hat{Y}_t; \phi))\mathrm{d}t + \sqrt{\beta_t}\mathrm{d}\bar{W}_t, \tag{3}$$

where $\hat{Y}_T \sim \pi = \mathcal{N}_d(0, I_d)$. The existing approach matches the score function $\nabla \log q_t(X_t)$ with an objective function (Hyvärinen, 2005; Song et al., 2020), which aims to learn the score function by minimizing

$$\mathcal{J}_{\mathrm{SM}}(\phi) := \frac{1}{2}\int_0^T \lambda(t)\, \mathbb{E}_{X_0 \sim q_0}\mathbb{E}_{X_t \sim q_{t|0}} \tag{4}$$

$$\left\|\nabla \log q_t(X_t|X_0) - s_t(X_t; \phi)\right\|^2 \mathrm{d}t + \tilde{C}_T,$$

where $\lambda(t)$ is a positive weighting function and $s_t(X_t; \phi)$ is a neural network (NN) with parameter $\phi$. The rationale for the approach is the following inequality (Corollary 1 in Song et al. (2021b)):

$$-\mathbb{E}_{X_0}\left[\log q_0(X_0; \phi)\right] \leq \mathcal{J}_{\mathrm{SM}}(\phi) + C_1, \tag{5}$$

where $C_1$ is a constant independent of $\phi$. This inequality explicitly shows that classical score matching only minimizes an upper bound on the negative log-likelihood rather than the likelihood itself. However, recent analyses (Koehler et al., 2023) have shown that this can lead to a severe loss of statistical efficiency compared to MLE, even for simple families of distributions like exponential families. Motivated by this limitation, we propose an approach that directly minimizes the negative log-likelihood $-\mathbb{E}_{X_0}[\log q_0(X_0; \phi)]$ instead of its upper bound $\mathcal{J}_{\text{SM}}$.

To derive the relationship between the likelihood of forward and backward trajectories, by a property of reversal diffusion (Haussmann & Pardoux, 1986), for any chosen time steps $0 = t_0 < t_1 < \cdots < t_{N-1} < t_N = T$, there is an equivalence of the joint likelihoods between the forward and the reverse processes:

$$q_{0:T}(x_0, x_1, \cdots, x_T; \theta) = p_{0:T}(x_0, x_1, \cdots, x_T; \theta), \quad (6)$$

where $q_{t_0:t_N}$ and $p_{t_0:t_N}$ represent the joint densities of the processes $\{X_{t_k}\}_{k=0}^N$ and $\{Y_{t_k}\}_{k=0}^N$.

The following proposition shows that the expected log-likelihood at $t = 0$ can be expressed by transition and marginal densities of the forward and the time-reversal processes. It will serve to construct the wanted likelihood approximation.

**Proposition 3.1.** *Suppose that there exists a positive constant $C$ such that $0 < \beta_t \leq C$ for any $t \in [0,T]$, and for any open bounded set $\mathcal{O} \subseteq \mathbb{R}^d$, $\int_0^T \int_{\mathcal{O}} (\|q_t(x; \theta)\|^2 + d \cdot \beta_t \|\nabla q_t(x; \theta)\|^2) \mathrm{d}x \mathrm{d}t < \infty$, then*

$$\mathbb{E}_{X_{t_0} \sim q_{t_0}} \log q_{t_0}(X_{t_0}; \theta) \quad (7)$$

$$= \mathbb{E}_{X_{t_0:t_N} \sim q_{t_0:t_N}} \left\{ \sum_{k=1}^N \log p_{t_{k-1}|t_k}(X_{t_{k-1}}|X_{t_k}; \theta) \right.$$

$$+ \log \underbrace{p_{t_N}(X_{t_N}; \theta)}_{\text{converge to } \mathcal{N}_d(0, I_d)} - \sum_{k=1}^N \log \underbrace{q_{t_k|t_{k-1}}(X_{t_k}|X_{t_{k-1}})}_{\text{given by (1) (independent of } \theta)} \left. \right\}$$

*for any $0 < t_1 < \cdots < t_{N-1} < T$.*

Proposition 3.1 links the expected log-likelihood of the initial distribution to that involving the forward process and the reverse process. As the forward transition density $q_{t_k|t_{k-1}}(X_{t_k}|X_{t_{k-1}})$ is free of the parameter $\theta$ due to $\beta_t$ being known, and for sufficiently large $t_N$, the density $p_{t_N}(X_{t_N}; \theta)$ converges to a stationary distribution $\mathcal{N}_d(0, I_d)$ that is also independent of $\theta$, (7) becomes

$$- \mathbb{E}_{X_0 \sim q_0}[\log q_0(X_0; \theta)]$$

$$\approx - \mathbb{E}_{X_{t_0:t_N} \sim q_{t_0:t_N}} \left\{ \sum_{k=1}^N \log p_{t_{k-1}|t_k}(X_{t_{k-1}}|X_{t_k}; \theta) \right\} + C_T$$

$$=: \mathcal{L}(\theta) + C_T, \quad (8)$$

where $C_T$ denotes a constant free of $\theta$.

The approximation in (8) arises from using a finite terminal time $T$ instead of infinity. This truncation error is well-controlled; as established in Appendix B (Lemma B.1), the KL divergence between the perturbed data distribution $q_T$ and the prior distribution converges to zero at a polynomial rate with respect to $T$. Expression (8) suggests a more attractive strategy, that is to minimize a computable version of $\mathcal{L}(\theta)$ rather than minimizing a version of the upper bound $\mathcal{J}_{\text{SM}}(\phi)$ in (5). In the next section, we detail an approach using the Quasi Maximum Likelihood, which allows constructing a tractable objective function by specifying an analytical form for these conditional log-likelihood terms.

Moreover, the arbitrariness of $t_1 < \cdots < t_{N-1}$ in (7) offers convenience for designing efficient algorithms to realize the approximation of $\mathcal{L}(\theta)$.

## 4. Methodology

We assume access to the original data $\{X_0^{(i)}\}_{i=1}^n$ where each $X_0^{(i)} \in \mathbb{R}^d$ at $t = 0$. For any fixed index $i$, we can generate a sequence of discrete observations $\{X_{t_k}^{(i)}\}_{k=0}^N$ according to the SDEs (1). Throughout the paper, we denote by $T > 0$ the diffusion horizon of the continuous-time SDE, and by $N$ the number of discrete reverse transition densities evaluated per path in the LM objective. In the theoretical analysis (Section 5) we set $t_k = k$ and $N = T$ with unit time increments, while in the experiments (Section 6) we draw a random grid of $N$ time points from $[0, T]$ as in Algorithm A.

### 4.1. Quasi-Maximum Likelihood Estimation

To handle the intractable $p_{t_{k-1}|t_k}$ in $\mathcal{L}(\theta)$, we adopt the Quasi-Maximum Likelihood approach (QML) (Wedderburn, 1974). This involves replacing the intractable true reverse transition density $p_{t_{k-1}|t_k}$ with a tractable proxy. Specifically, we use a Gaussian distribution whose mean and covariance match the true conditional mean and covariance of the reverse process, which are derived in Proposition 4.1. As $Y_t \overset{d}{=} X_t$ and the joint PDF equivalence (6), these moments are the same as those of $q_{t_{k-1}|t_k}(X_{t_{k-1}}|X_{t_k}; \theta)$. As shown in Appendix B, the true reverse transition density can be written as the matched Gaussian reference density multiplied by a higher-order exponential remainder. Thus, the misspecification induced by QMLE is controlled by the discretization error and vanishes as the time step decreases.

The following proposition provides the analytical forms of these conditional mean and covariance, which are used to define matched Gaussian distribution in the quasi-likelihood.

**Proposition 4.1.** *Let $\mu_{s|t}$ and $\Sigma_{s|t}$ be the conditional mean and covariance of $q_{s|t}(X_s|X_t; \theta)$, respectively, for $s < t$.*

*Then,*

$$\mu_{s|t} = \mathbb{E}\left(X_s|X_t\right) = \frac{X_t + \sigma_{t|s}^2 \nabla \log q_t(X_t;\theta)}{m_{t|s}} \quad and$$

$$\Sigma_{s|t} = \mathbb{E}\left[\left(X_s - \mu_{s|t}\right)\left(X_s - \mu_{s|t}\right)^T |X_t\right]$$
$$= \frac{\sigma_{t|s}^2}{m_{t|s}^2}\left(I_d + \sigma_{t|s}^2 \nabla^2 \log q_t(X_t;\theta)\right),$$

*where* $m_{t|s} = \exp\{-\int_s^t \beta_t \mathrm{d}t/2\}$ *and* $\sigma_{t|s}^2 = 1 - \exp\{-\int_s^t \beta_t \mathrm{d}t\}$.

To facilitate the QMLE approach, we parameterize both $\nabla \log q_t(X_t;\theta)$ and the Hessian function $\nabla^2 \log q_t(X_t;\theta)$. This parameterization strategy adapts to whether the data's parametric family $\mathcal{F}_\theta$ is known a priori. In specialized domains like financial modeling or signal processing, where $\mathcal{F}_\theta$ can be known, these functions can be expressed analytically in terms of the true parameters $\theta$, a property we use for parameter estimation in Section 6.1. More commonly, for complex high-dimensional data like images where $\mathcal{F}_\theta$ is unknown, we employ neural networks as universal approximators. Our implementation uses two separate U-Net models to represent the score $s_t(x;\phi)$ and the Hessian $H_t(x;\phi)$, where $\phi$ denotes their learnable parameters.

The quasi-likelihood approximation to transition density $p_{t_{k-1}|t_k}(Y_{t_{k-1}}|Y_{t_k};\phi)$ is

$$\hat{p}_{t_{k-1}|t_k}(Y_{t_{k-1}}|Y_{t_k};\phi)$$
$$= \varphi_d(Y_{t_{k-1}};\mu_{t_{k-1}|t_k}(Y_{t_k};\phi),\Sigma_{t_{k-1}|t_k}(Y_{t_k};\phi)), \quad (9)$$

where $\varphi_d(x;\mu,\Sigma)$ denotes the $d$-dimensional Gaussian density. The matched conditional mean and covariance are given by

$$\mu_{t_{k-1}|t_k}(Y_{t_k};\phi) = \frac{1}{m_{t_k|t_{k-1}}}\left(Y_{t_k} + \sigma_{t_k|t_{k-1}}^2 s_{t_k}(Y_{t_k};\phi)\right),$$

$$\Sigma_{t_{k-1}|t_k}(Y_{t_k};\phi) = \frac{\sigma_{t_k|t_{k-1}}^2}{m_{t_k|t_{k-1}}^2}\left\{I_d + \sigma_{t_k|t_{k-1}}^2 H_{t_k}(Y_{t_k};\phi)\right\}.$$

With the quasi-Gaussian specification (9), we define the population-level quasi-log-likelihood objective function

$$\mathcal{L}(\phi) = -\sum_{k=1}^N \mathbb{E}_{X_{t_0:t_N}\sim q_{t_0:t_N}}\left\{\log \hat{p}_{t_{k-1}|t_k}(X_{t_{k-1}}|X_{t_k};\phi)\right\} \quad (10)$$

based on the forward data processes by noting (7) and (9).

Let $\ell_{\{t_0,\cdots,t_N\}}^{(i)}(\phi) = -\sum_{k=1}^N \log \hat{p}_{t_{k-1}|t_k}(X_{t_{k-1}}^{(i)}|X_{t_k}^{(i)};\phi)$, where $X_{t_k}^{(i)} = m_{t_k|t_{k-1}}X_{t_{k-1}}^{(i)} + \sigma_{t_k|t_{k-1}}Z_{t_k}^{(i)}$ be the realized path of the forward SDE (1) and $\{Z_{t_k}^{(i)}\}_{k=1}^N$ are IID standard Gaussian noise, and let

$$\mathcal{J}_{n,N}(\phi) = n^{-1}\sum_{i=1}^n \ell_{\{t_0,\cdots,t_N\}}^{(i)}(\phi) \quad (11)$$

be the aggregated sample quasi-log-likelihood, which depends on the choices of $\{t_0,t_1,\cdots,t_N\}$. Let $\hat{\phi}_{n,N} = \arg\min_\phi \mathcal{J}_{n,N}(\phi)$ be the quasi-MLE. Substituting $s_t(Y_t;\hat{\phi}_{n,N})$ to the reverse SDE (2) yields the modified reverse SDE

$$\mathrm{d}\hat{Y}_t = \frac{1}{2}\beta_t(\hat{Y}_t + 2s_t(\hat{Y}_t;\hat{\phi}_{n,N}))\mathrm{d}t + \sqrt{\beta_t}\mathrm{d}\bar{W}_t,$$

where $\hat{Y}_T \sim \pi = \mathcal{N}_d(0,I_d)$ and denote the density of $\hat{Y}_t(\hat{\phi}_{n,N})$ by $p_t(\cdot;\hat{\phi}_{n,N})$. For notational simplicity, in the rest of this paper, we use $q_t \equiv q_t(\cdot;\theta)$, $\hat{p}_t \equiv p_t(\cdot;\hat{\phi}_{n,N})$, $\hat{s}_t(\cdot) \equiv s_t(\cdot;\hat{\phi}_{n,N})$ and $\hat{H}_t(\cdot) \equiv H_t(\cdot;\hat{\phi}_{n,N})$.

**Stochastic Sampler.** Proposition 4.1 implies the following sampling procedure that differs from the conventional DDPM-type sampler (Ho et al., 2020):

$$\tilde{Y}_{t-1} = \hat{\mu}_{t-1|t}(\tilde{Y}_t) + \widehat{\Sigma}_{t-1|t}^{\frac{1}{2}}(\tilde{Y}_t)Z_t \quad (12)$$

for $t = T,\cdots,1$, where $Z_t \stackrel{\mathrm{IID}}{\sim} \mathcal{N}_d(0,I_d)$ and

$$\hat{\mu}_{t-1|t}(\tilde{Y}_t) = m_{t|t-1}^{-1}(\tilde{Y}_t + \sigma_{t|t-1}^2 \hat{s}_t(\tilde{Y}_t)),$$
$$\widehat{\Sigma}_{t-1|t}(\tilde{Y}_t) = m_{t|t-1}^{-2}\sigma_{t|t-1}^2\{I_d + \sigma_{t|t-1}^2 \hat{H}_t(\tilde{Y}_t)\}, \quad (13)$$

which involves the score and the Hessian function. Similarly, we denote the PDF of $\tilde{Y}_t$ generated by (12) as $\tilde{p}_t$. An efficient implementation of the sampling scheme is given in Appendix C.5.

### 4.2. Likelihood Matching and Efficient Algorithms

To realize the Quasi-Likelihood (11), the intermediate time points $t_1$ through $t_{N-1}$ are fixed in advance. To effectively utilize the evolutionary information from the forward SDEs, practitioners often employ an exceedingly large number of discretization steps, say $N$, to generate training data. However, such fine-grained discretization imposes significant computational burdens on training both the score model $s_t$ and the Hessian model $H_t$. To address this issue, we propose a more efficient computational algorithm.

We note that Proposition 3.1 holds true for arbitrary time points $0 < t_1 < \cdots < t_{N-1} < T$, which enables a time-averaged version of (10), expressed as:

$$-\frac{(N-1)!}{T^{N-1}}\int_0^T \cdots \int_0^{t_2}$$
$$\sum_{k=1}^N \mathbb{E}_{X_{t_0:t_N}\sim q_{t_0:t_N}}\left\{\log \hat{p}_{t_{k-1}|t_k}(X_{t_{k-1}}|X_{t_k};\phi)\right\}$$
$$\mathrm{d}t_1 \cdots \mathrm{d}t_{N-1}.$$

An empirical version can be constructed as follows. At the $r$-th stochastic gradient descent (SGD) iteration, for each

training trajectory $i$, we independently sample an ordered time grid $(t_1^{(i,r)}, \cdots, t_{N-1}^{(i,r)})$ from the uniform distribution over the simplex

$$\mathcal{T} = \{(t_1, \cdots, t_{N-1}) \in (0,T)^{N-1} : t_1 < \cdots < t_{N-1}\}.$$

We set $t_0^{(i,r)} = 0$ and $t_N^{(i,r)} = T$. For compactness, write

$$\tau_k^{ir} = t_k^{(i,r)}, \quad X_k^{ir} = X_{\tau_k^{ir}}^{(i)}, \quad \text{and } \boldsymbol{\tau}^{ir} = (\tau_0^{ir}, \ldots, \tau_N^{ir}),$$

where $\tau_0^{ir} = 0$, $\tau_N^{ir} = T$, and the superscript $ir$ is a compact label for the pair $(i,r)$. This yields the following stochastic approximation to the time-averaged LM objective:

$$\tilde{J}_{n,N}^{(r)}(\phi) = \frac{1}{n}\sum_{i=1}^{n} \ell_{\boldsymbol{\tau}^{ir}}^{(i)}(\phi) = \frac{1}{n}\sum_{i=1}^{n}\sum_{k=1}^{N} \ell_k^{ir}(\phi), \qquad (14)$$

where

$$\ell_k^{ir}(\phi) = -\log \hat{p}_{\tau_{k-1}^{ir}|\tau_k^{ir}}(X_{k-1}^{ir} \mid X_k^{ir}; \phi) \qquad (15)$$

$$= \frac{1}{2}\log|\Sigma_k^{ir}(\phi)| + \frac{1}{2}\left\|\Sigma_k^{ir}(\phi)^{-\frac{1}{2}}(X_{k-1}^{ir} - \mu_k^{ir}(\phi))\right\|^2,$$

with

$$\mu_k^{ir}(\phi) = \mu_{\tau_{k-1}^{ir}|\tau_k^{ir}}(X_k^{ir}; \phi),$$
$$\Sigma_k^{ir}(\phi) = \Sigma_{\tau_{k-1}^{ir}|\tau_k^{ir}}(X_k^{ir}; \phi).$$

Here constants independent of $\phi$ are omitted. We call (14) the Likelihood Matching (LM) objective. The random time-point selection strategy allows a more comprehensive temporal evaluation during training, even with a modest transition step $N$ (see Appendix D for discussions).

Furthermore, by expanding equation (15), the second term in (15) becomes

$$\frac{1}{2}\left\|\left(I_d + \sigma_{\tau_k^{ir}|\tau_{k-1}^{ir}}^2 H_k^{ir}(\phi)\right)^{-1/2}\left(Z_k^{ir} + \sigma_{\tau_k^{ir}|\tau_{k-1}^{ir}} s_k^{ir}(\phi)\right)\right\|^2,$$

where $Z_k^{ir} = Z_{\tau_k^{ir}}^{(i)}$, $s_k^{ir}(\phi) = s_{\tau_k^{ir}}(X_k^{ir}; \phi)$ and $H_k^{ir}(\phi) = H_{\tau_k^{ir}}(X_k^{ir}; \phi)$. This expression unifies the score matching (Song et al., 2021c) and likelihood weighting (Song et al., 2021b) as special cases of the transition probability within our LM objective. In particular, when $\hat{H}_t \equiv 0$ the second term reduces to a rescaled $\ell_2$ score-matching loss

$$\frac{1}{2}\left\|Z_k^{ir} + \sigma_{\tau_k^{ir}|\tau_{k-1}^{ir}} s_k^{ir}(\phi)\right\|^2,$$

recovering standard score matching; the presence of $(I + \sigma^2 H_t)^{-1/2}$ plays the role of an automatically learned likelihood weight, while the extra $\log|\Sigma_{t_{k-1}|t_k}|$ term completes a proper quasi likelihood for the reverse transition. However, our formulation integrates covariance to weight the score while leveraging additional transition probabilities, thereby utilizing more trajectory information. The algorithm for Likelihood Matching is provided in Appendix A.

The LM objective (14) incorporates both score matching and an additional covariance matching. Moreover, it naturally weights each time step via the matched covariance, rather than relying on pre-specified weights, for instance, $\lambda(t)$ in (4). The experimental section analyzes how different number of generated time points $N$ in (14) affects the performance.

Exploiting the intrinsic dimensionality of real data distributions, Meng et al. (2021) proposed parameterizing $H_t(X_t; \phi)$ with low-rank matrices defined as $H_t(X_t; \phi) = \boldsymbol{U}_t(X_t; \phi) + \boldsymbol{V}_t(X_t; \phi)\boldsymbol{V}_t(X_t; \phi)^T$ where $\boldsymbol{U}_t(\cdot; \phi) : \mathbb{R}^d \to \mathbb{R}^{d \times d}$ is a diagonal matrix, and $\boldsymbol{V}_t(\cdot; \phi) : \mathbb{R}^d \to \mathbb{R}^{d \times r}$ is a matrix with a prespecified rank $r \ll d$ for a pre-determined $r$, reducing computational complexity. Beyond computational savings, the diagonal-plus-low-rank form acts as a structural regularizer motivated by the manifold hypothesis; its approximation error is absorbed into $\varepsilon_H$ in Assumption 5.3, and empirically $r \in [10, 30]$ gives the best quality–cost trade-off.

To efficiently compute the likelihood (14), we apply the Sherman-Morrison-Woodbury (SMW) formula, namely after suppressing argument $(X_t; \phi)$ in related quantities, for any $X \in \mathbb{R}^d$,

$$X^T(I_d + \sigma_{t_k|t_{k-1}}^2 \boldsymbol{U}_t + \sigma_{t_k|t_{k-1}}^2 \boldsymbol{V}_t\boldsymbol{V}_t^T)^{-1}X$$
$$= \tilde{X}^T\tilde{X} - (\tilde{\boldsymbol{V}}_t^T\tilde{X})^T(I_r + \tilde{\boldsymbol{V}}_t^T\tilde{\boldsymbol{V}}_t)^{-1}(\tilde{\boldsymbol{V}}_t^T\tilde{X}),$$

where $\tilde{X} = (I_d + \sigma_{t_k|t_{k-1}}^2 \boldsymbol{U}_t)^{-1/2}X$ and $\tilde{\boldsymbol{V}}_t = \sigma_{t_k|t_{k-1}}(I_d + \sigma_{t_k|t_{k-1}}^2 \boldsymbol{U}_t)^{-1/2}\boldsymbol{V}_t$. Similarly, the determinant can be computed efficiently using the matrix determinant lemma:

$$|I_d + \sigma_{t_k|t_{k-1}}^2 \boldsymbol{U}_t + \sigma_{t_k|t_{k-1}}^2 \boldsymbol{V}_t\boldsymbol{V}_t^T|$$
$$= |I_d + \sigma_{t_k|t_{k-1}}^2 \boldsymbol{U}_t| \cdot |I_r + \tilde{\boldsymbol{V}}_t^T\tilde{\boldsymbol{V}}_t|.$$

More details are in the Appendix C.5.

## 5. Theoretical Analysis

In the theoretical analysis, we set $t_k = k$ and $N = T$, and specify the noise schedule $\beta_t$ similar to Li et al. (2024b) (details in Appendix B), and assume the following assumptions.

**Assumption 5.1** (Boundedness of the Distribution). The original data distribution $q_0$ possesses a bounded second-order moment such that $\mathbb{E}_{X \sim q_0}\|X\|^2 \leq M_2$ for a positive constant $M_2$.

**Assumption 5.2** ($L_2$ Score Estimation Error). The estimated score function $\hat{s}_t(x)$ satisfies $T^{-1}\sum_{t=1}^{T}\mathbb{E}_{X \sim q_t}\|\nabla \log q_t(X) - \hat{s}_t(X)\|^2 \leq \varepsilon_s^2$ for a constant $\varepsilon_s > 0$.

**Assumption 5.3** (Frobenius Hessian Estimation Error). The estimated Hessian function $\hat{H}_t(x)$ satisfies $T^{-1}\sum_{t=1}^{T}\mathbb{E}_{X\sim q_t}\|\nabla^2\log q_t(X) - \hat{H}_t(X)\|_F^2 \leq \varepsilon_H^2$ for a constant $\varepsilon_H > 0$.

**Assumption 5.4.** For $t = 1,\cdots,T$, the true Hessian function $\nabla^2\log q_t(x)$ satisfies $\lambda_{\min}((1-\alpha_t)\nabla^2\log q_t(x)) \geq \varepsilon_0 > -1$, where $\varepsilon_0$ is constant.

Assumption 5.1-5.3 are standard in the literature (Li et al., 2024b;a; Benton et al., 2023; Chen et al., 2023). Assumption 5.4 is a discrete-time non-degeneracy condition ensuring that the matched covariance remains well-conditioned. By Proposition 4.1, we know that $I_d + (1-\alpha_t)\nabla^2\log q_t(x) \succeq 0$, so the eigenvalues of $(1-\alpha_t)\nabla^2\log q_t(x)$ are lower bounded by $-1$. Empirically, Gaussian smoothing quickly improves conditioning and we provide supporting eigenvalue diagnostics in Table 2 in Appendix C.

**Theorem 5.5** (Non-asymptotic Bound for Distributions with Bounded Moments). *Under Assumptions 5.1-5.4, the generated distribution $\tilde{p}_0$ by Sampler (12) satisfies*

$$\mathrm{TV}(q_0\|\tilde{p}_0) \leq \sqrt{\frac{1}{2}\mathrm{KL}(q_0\|\tilde{p}_0)}$$
$$\lesssim \frac{d^3\log^{4.5}T}{T} + \sqrt{\log T}\varepsilon_s + \frac{\log T}{\sqrt{T}}\varepsilon_H.$$

Theorem 5.5 provides non-asymptotic convergence guarantees for the stochastic sampler (12). The error bound consists of three terms: the reverse step error that scales as $O(d^3\log^{4.5}T/T)$, reflecting the discrepancy between forward and reverse transition densities; the score estimation error and Hessian estimation errors which are $O(\sqrt{\log T})$ and $O(\log T/\sqrt{T})$, respectively, due to utilization of mean and covariance information in the sampling procedure. To achieve the $\varepsilon$-accuracy approximation error, assuming the exact score, the total number of time steps $T$ should be $O(d^3/\varepsilon)$.

In the following theorem, we consider an oracle parametric setting where the score can be written as $s_t(x;\theta) = \nabla\log q_t(x;\theta)$ for a finite-dimensional parameter $\theta$. Thus, unlike the nonparametric neural-network setting where we denote network weights by $\phi$, here $\theta$ directly indexes the data-generating family $\mathcal{F}_\theta$.

**Theorem 5.6** (Consistency under Oracle Model). *Suppose $s_t(x;\theta) = \nabla\log q_t(x;\theta)$ for any $t \geq 0$. Then, under conditions given in the Appendix B.4, the quasi-MLE $\hat{\theta}_{n,T} \xrightarrow{p} \theta^*$ in probability as $n,T \to \infty$, where $\theta^*$ is the parameter of the original data distribution $q_0$.*

The theorem shows that when the true form of the score function is accessible, the estimation by minimizing the Likelihood Matching objective (14) converges to the true value.

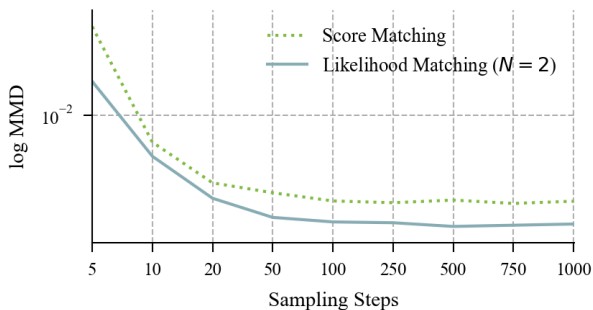

*Figure 2.* Highly coiled Swiss-roll experiment. We report log-MMD between generated and true samples as the number of sampling steps varies. LM with $N = 2$ consistently improves over SM, supporting the benefit of Hessian-informed covariance on curved probability landscapes.

# 6. Experiments

This section reports empirical results to validate our theory and methodological insights through numerical experiments on both synthetic datasets and image datasets. To ensure the reproducibility of our results, we provide a comprehensive description of all experimental details, including experiment setting, additional results, and an analysis of the computational time and memory consumption, in Appendix C.

## 6.1. Synthetic Datasets

**Mixture Model.** To analyze a known failure case of Score Matching, which can struggle to accurately fit mixture distributions with well-separated modes (Koehler et al., 2023), we first considered a two-component Gaussian mixture with equal weights and means located at -10 and 10, with unit variance. We also examined mixtures of the $t$-distributions with 3 degrees of freedom under the same settings. We used a single -hidden-layer multilayer perceptron (MLP) to model both the score and Hessian functions, and compared the performance of the Score Matching method with the Likelihood Matching method using transition step $N = 2$, 3, and 8. For evaluation, we used the Maximum Mean Discrepancy (MMD, Gretton et al. 2012), which employed five Gaussian kernels with bandwidths $\{2^{-2}, 2^{-1}, 2^0, 2^1, 2^2\}$. The results, averaged over 100 independent trials, are reported in Figures 3 (a)-(b), which show that the proposed LM consistently outperformed the Score Matching, and the performance of the LM improved as $N$ increases.

To further test whether Hessian-informed covariance helps on curved probability landscapes, we considered a highly coiled two-dimensional Swiss-roll distribution. This setting complements the separated mixtures because its mass concentrates near a curved manifold, where first-order reverse approximations can be less accurate. We trained LM

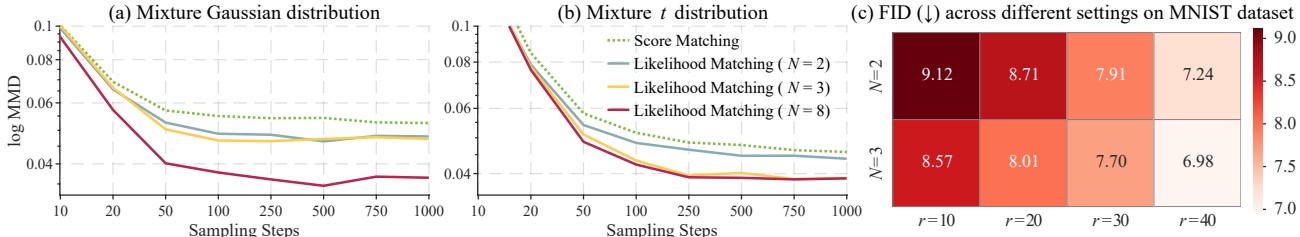

*Figure 3.* Maximum Mean Discrepancy (MMD; lower is better) between generated and true samples under two 1D mixture distributions: (a) Gaussian and (b) $t$ with 3 degrees of freedom with respect to the number of sampling steps $N$. (c) Fréchet Inception Distance (FID; lower is better) on the MNIST dataset for different combinations of $(N, r)$ under the Likelihood Matching framework.

with $N = 2$ and compared it with SM under the same sampling budget. As shown in Figure 2, LM achieved consistently lower log-MMD across sampling steps, supporting the role of matched covariance in high-curvature distributions. Because the ground-truth density is available in the 1D Gaussian mixture, Appendix C.3 further reports held-out NLL and $KL(q_0 \| \hat{p}_0)$, where LM improves over SM under likelihood-oriented metrics as well as MMD.

**Parameter Estimation.** We evaluated the LM approach on a two-dimensional Gaussian mixture distribution, i.e., $q_0(x) \sim \omega_1 \mathcal{N}_2(\mu_1, \sigma_1^2 I_2) + (1 - \omega_1) \mathcal{N}_2(\mu_2, \sigma_2^2 I_2)$, where the score model (with the true oracle score) can be derived analytically. Using ground truth parameters $\mu_1 = (1, 2)^T$, $\mu_2 = (-1, -3)^T$, $\sigma_1 = \sqrt{0.3}$, $\sigma_2 = \sqrt{0.6}$ and $\omega_1 = 1/3$, we compared parameter estimation between the Likelihood Matching (LM) and Score Matching methods. For sample sizes $n = 100$ and 200 (500 replicates each), we report the mean absolute error (MAE) and standard error (Std. Error) in Table 4 (Appendix C.3), which showed that the LM had consistently lower MAE and Std. Error than the Score Matching. The decreasing estimation variance of the LM with increasing sample size supported the consistency guarantee in Theorem 5.6.

### 6.2. Image Datasets

The performance of the Likelihood Matching is expected to improve as both $N$ and $r$ increase. To verify this, we trained the Likelihood Matching model on the MNIST dataset under different settings of $(N, r)$. The FID (Fréchet Inception Distance) for each setting is presented in Figure 3 (c), which aligns well with the expectation.

We evaluated our LM framework on the CIFAR-10 and CelebA 64x64 datasets, comparing it against an SM baseline. As shown in Table 1, the LM method consistently outperforms SM across all metrics. Notably, even with a simple diagonal Hessian approximation ($r = 0$), LM achieves a lower FID on both CIFAR-10 (3.12 vs. 3.15) and CelebA (2.69 vs. 2.71), alongside improved negative log-likelihood (NLL) where the NLL metric is computed directly in the

*Table 1.* Quantitative comparison on CIFAR-10 and CelebA 64x64. LM with fixed transition steps ($N = 2$) and varying Hessian ranks ($r$) is compared against the Score Matching (SM) baseline. FID ($\downarrow$) and NLL ($\downarrow$) indicate lower is better, while IS ($\uparrow$) indicates higher is better.

| | CIFAR10 FID $\downarrow$ | CIFAR10 IS $\uparrow$ | CIFAR10 NLL (bpd) $\downarrow$ | CelebA $64 \times 64$ FID $\downarrow$ |
|---|---|---|---|---|
| SM | 3.15 | $9.47 \pm 0.10$ | 3.28 | 2.71 |
| LM ($r = 0$) | 3.12 | $9.47 \pm 0.11$ | 3.24 | 2.69 |
| LM ($r = 10$) | 3.04 | $9.46 \pm 0.13$ | 3.15 | 2.67 |
| LM ($r = 20$) | **3.01** | **9.48 $\pm$ 0.14** | 3.13 | 2.65 |
| LM ($r = 30$) | 3.03 | $9.45 \pm 0.13$ | **3.11** | **2.62** |
| LM ($r = 100$) | 3.05 | $9.46 \pm 0.12$ | 3.12 | 2.63 |
| LM ($r = 200$) | 3.09 | $9.46 \pm 0.15$ | 3.13 | 2.64 |

discrete SDE formulation by evaluating the exact Gaussian likelihood of the residuals under the learned covariance, as detailed in Appendix C.

We note that the NLL values in Table 1 are evaluated within our discrete-SDE quasi-Gaussian formulation by computing the Gaussian likelihood of reverse residuals under the learned covariance. Under this metric, LM achieves 3.11 bpd on CIFAR-10, compared with 3.28 bpd for the SM baseline. This is not directly identical to the likelihood objectives used by continuous-time ODE likelihood methods or variational diffusion models. For context, DDPM reports 3.70 bpd on CIFAR-10, while likelihood-oriented score-based ODE training reports 3.66 bpd for the VE baseline and 3.44 bpd for its second-order variant; these numbers should be interpreted cautiously because the likelihood decompositions and evaluation pipelines differ.

The performance gains increase with the Hessian rank, peaking around $r = 20 - 30$. This demonstrates a clear benefit from incorporating covariance information. For instance, LM with $r = 30$ achieves a FID of 3.03 on CIFAR-10 and 2.62 on CelebA, a notable improvement over the SM baseline. These results provide strong empirical evidence that our likelihood-based objective is more effective than score matching for training high-fidelity generative models.

We then performed sampling on the MNIST dataset using the sampler described in (12) based on the trained Likelihood Matching ($N = 2$, $r = 10$) and the Score Matching, where the Hessian function in Score Matching is fixed as

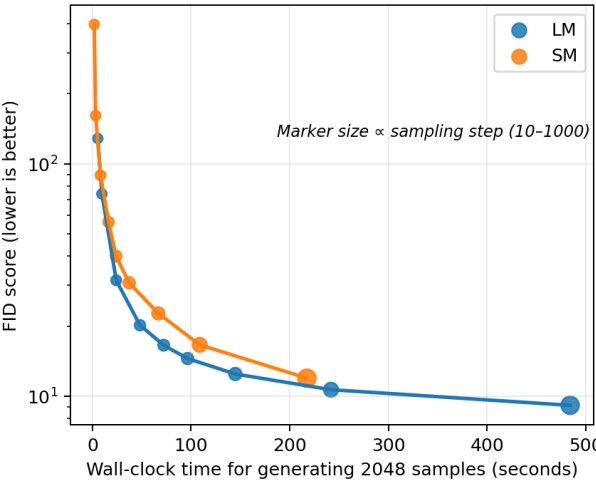

*Figure 4.* Speed–quality trade-off on MNIST. We compare LM ($N = 2, r = 10$) and the SM baseline by varying the number of reverse sampling steps from 10 to 1000. The marker size is proportional to the sampling step. LM achieves lower FID under comparable wall-clock time, with the largest gain in the low-step regime.

zero. Both methods perform well with a large number of sampling steps. Figure 8 (Appendix C.3) presents the results under fewer sampling steps, where we observe that Likelihood Matching exhibits faster convergence and better preservation of the structural integrity of the generated images. Qualitatively, after only 20 reverse iterations the LM sampler already produces clearly recognizable digits, whereas the corresponding SM samples remain noticeably more blurred and less structured, indicating that the Hessian-based covariance improves the per-step accuracy of the reverse transition and effectively reduces the number of sampling steps needed to reach a given visual quality.

A more detailed analysis of training and sampling time, as well as GPU memory usage, is given in Appendix C. In particular, Table 5 reports the overhead on CIFAR-10, while Table 6 shows that on $224 \times 224$ ImageNet our method increases training time by roughly $3$–$4\times$ and memory by about $2$–$3\times$ over the SM baseline, confirming both the scalability challenge and the current computational feasibility of LM.

Figure 4 further reported a speed–quality Pareto comparison. Although LM had a higher per-step cost due to Hessian modeling, it achieved lower FID for comparable wall-clock time and reached a given quality level with fewer reverse steps, especially in the low-step regime.

Appendix C further includes a class-conditional MNIST experiment, showing that LM can be used with the standard conditional score-network interface without modifying the conditioning mechanism.

## 6.3. Ablation Studies

**Marginal Benefit of Hessian.** To isolate the contribution of the learned Hessian, we conducted an ablation study on MNIST where the score network was trained but the Hessian was set to a fixed identity matrix $H_t \equiv I$. This score-only LM variant resulted in significantly worse FID scores (10.28 for $N = 2$ and 9.75 for $N = 3$) compared to the full LM model, confirming that explicitly modeling the covariance is crucial for performance. Empirically, we find that relatively small ranks yield the best trade-off between performance and computational cost. As shown in Table 1, moving from a diagonal Hessian ($r = 0$) to $r = 20 - 30$ brings noticeable gains, while higher ranks ($r = 100, 200$) offer diminishing returns. Therefore, we recommend $r \in [10, 30]$ as a practical guideline for standard image benchmarks.

## 7. Conclusion

This work introduces the Likelihood Matching method for training diffusion models, which is grounded in the Maximum Likelihood Estimation, by leveraging on the Quasi-Maximum Likelihood Estimation (QMLE). The approach inherently integrates both score and covariance matching, distinguishing it from the score matching that focuses solely on a single transition density and utilizes only the first-order moment information. Our theoretical analysis establishes the consistency of the QMLE and provides non-asymptotic convergence guarantees for the proposed sampler quantifying the impact of the score and Hessian estimation errors, dimensionality, and diffusion steps. Empirical evaluations on image datasets demonstrated the viability of the proposed approach and elucidated the influence of methodological choices such as Hessian approximation rank. Our comprehensive evaluations show that LM consistently outperforms the foundational SM baseline in generation quality and likelihood estimation, with a manageable and scalable increase in computational cost.

Building upon this robust foundation, future directions involve exploring the application of our methods to more challenging, high-dimensional data domains, such as high-resolution natural images or video generation. Concurrently, enhancing the computational efficiency of both the training and sampling procedures represents another promising avenue for further research, aimed at broadening the practical applicability of LM. In particular, the computational burden of Hessian modeling on large-scale datasets remains substantial, and alleviating this limitation is an important direction for future work. Our LM objective is also complementary to the rich body of work on optimal covariance design and non-Gaussian transition approximations in diffusion models, and combining LM with such advanced solvers on large-scale benchmarks such as ImageNet is an especially promising direction.

## Acknowledgements

This research was supported by the Fundamental and Interdisciplinary Disciplines Breakthrough Plan of the Ministry of Education of China Grant No. JYB2025XDXM801, and the National Natural Science Foundation of China Grant Nos. 12292980, 12292983. We thank for the technical support of the National Large Scientific and Technological Infrastructure "Earth System Numerical Simulation Facility". We also thank the anonymous reviewers for their valuable feedback.

## Impact Statement

This paper presents work whose goal is to advance the field of Generative Learning. There are many potential societal consequences of our work, none which we feel must be specifically highlighted here.

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

# A. Algorithm

---

**Algorithm 1** Likelihood Matching without time random sampling

---

**Input:** Dataset $\mathcal{D} = \{X_0^{(i)}\}_{i=1}^n \overset{\text{IID}}{\sim} q_0$, pre-determined time step set $\mathcal{T} = \{t_k\}_{k=1}^{N-1}$, learning rate $\eta$, batch size $B$

**Output:** Trained model: $\hat{s}_t(X_t; \phi)$ and $\hat{H}_t(X_t; \phi)$

// Training

**while** not converge **do**

  **for** each batch **do**

    Get a mini-batch $\{X_0^{(i)}\}_{i=1}^B$ from $\mathcal{D}$ ($X_0^{(i)}$ is the $i$-th sample in the current batch)

    **for** $k = 1, 2, \cdots, N$ **do**

      Get perturbed data $\{X_t^{(i)}\}_{t=t_0}^{t_N}$

      Calculate the transition term $\log \hat{p}_{t_{k-1}|t_k}(X_{t_{k-1}}^{(i)} | X_{t_k}^{(i)}; \phi)$

    **end for**

    Calculate the batch loss $\mathcal{L}(\theta) = -B^{-1} \sum_{i=1}^B \sum_{k=1}^N \log \hat{p}_{t_{k-1}|t_k}(X_{t_{k-1}}^{(i)} | X_{t_k}^{(i)}; \phi)$

    Update the parameter of $s_t$ and $H_t$ via SGD on:

$$\phi \leftarrow \phi - \eta \nabla_\phi \mathcal{L}(\phi)$$

  **end for**

**end while**

Obtain the trained $\hat{s}_t(X_t; \phi)$ and $\hat{H}_t(X_t; \phi)$ // Also obtain the estimated parameter $\hat{\phi}$

---

**Algorithm 2** Likelihood Matching with time random sampling

---

**Input:** $\mathcal{D} = \{X_0^{(i)}\}_{i=1}^n \overset{\text{IID}}{\sim} q_0$, learning rate $\eta$, batch size $B$, the number of chosen time points $N$

**Output:** Trained model: $\hat{s}_t(X_t; \phi)$ and $\hat{H}_t(X_t; \phi)$ // Training

**while** not converge **do**

  **for** each batch **do**

    Get a mini-batch $\{X_0^{(i)}\}_{i=1}^B$ from $\mathcal{D}$ ($X_0^{(i)}$ is the $i$-th sample in the current batch)

    Sample $(t_1^{(r)}, \cdots, t_{N-1}^{(r)}) \sim \text{Unif}\{(0, T)\}$ where $t_1^{(r)} < \cdots < t_{N-1}^{(r)}$ for $i = 1, \cdots, B$

    **for** $k = 1, 2, \cdots, N$ **do**

      Get perturbed data $\{X_t^{(i)}\}_{t=t_0}^{t_N}$

      Calculate the transition term $\log \hat{p}_{t_{k-1}^{(r)}|t_k^{(r)}}(X_{t_{k-1}^{(r)}}^{(i)} | X_{t_k^{(r)}}^{(i)}; \phi)$

    **end for**

    Calculate the batch loss $\mathcal{L}(\theta) = -B^{-1} \sum_{i=1}^B \sum_{k=1}^N \log \hat{p}_{t_{k-1}^{(r)}|t_k^{(r)}}(X_{t_{k-1}^{(r)}}^{(i)} | X_{t_k^{(r)}}^{(i)}; \phi)$

    Update the parameter of $s_t$ and $H_t$ via SGD on:

$$\phi \leftarrow \phi - \eta \nabla_\phi \mathcal{L}(\phi)$$

  **end for**

**end while**

Obtain the trained $\hat{s}_t(X_t; \phi)$ and $\hat{H}_t(X_t; \phi)$ // Also obtain the estimated parameter $\hat{\phi}$

---

# B. Technical Results and Proofs

We first review the noise schedule proposed in Li et al. (2024b). For sufficiently large constants $c_0, c_1 > 0$, define

$$e^{-\int_0^1 \beta_t \mathrm{d}t} = \alpha_1 = \frac{1}{T^{c_0}},$$
$$e^{-\int_{t-1}^t \beta_t \mathrm{d}t} = \alpha_t = \frac{c_1 \log T}{T} \min\{(1 + \alpha_1)(1 + \frac{c_1 \log T}{T})^t, 1\}. \tag{16}$$

As established by (Li et al., 2024b), this specification ensures $\alpha_t \geq 1/2$ and $1 - \alpha_t \lesssim \log T/T$.

## B.1. Proof of Proposition 3.1

According to the definition of a time-reversal process in (Haussmann & Pardoux, 1986), when $\beta_t$ of (1) is bounded and $\int_0^T \int_\mathcal{O} (\|q_t(x;\theta)\|^2 + d \cdot \beta_t \|\nabla q_t(x;\theta)\|^2) \mathrm{d}x \mathrm{d}t < \infty$, the time-reversal process of $X_t$ exits, i.e., we have $Y_t \overset{d}{=} X_t$ and $Y_t$ evolves from (2). Then the finite-dimensional distribution for the process $Y_t$ is identically distributed as the associated distribution for process $X_t$. Therefore, we have

$$q_{t_0:t_N}(x_{t_0}, x_{t_1}, \cdots, x_{t_N}; \theta) = p_{t_0:t_N}(x_{t_0}, x_{t_1}, \cdots, x_{t_N}; \theta), \tag{17}$$

for every $(x_{t_0}, x_{t_1}, \cdots, x_{t_N}; \theta) \in \mathbb{R}^{N+1} \times \Theta$. Thus, if we take the logarithm of both sides of the above equation, we will have

$$\log q_{t_0}(X_{t_0}; \theta) + \sum_{k=1}^N \log q_{t_k|t_{k-1}}(X_{t_k}|X_{t_{k-1}}) = \log p_{t_N}(X_{t_N}; \theta) + \sum_{k=1}^N \log p_{t_{k-1}|t_k}(X_{t_{k-1}}|X_{t_k}; \theta). \tag{18}$$

Taking the expectation with respect to $(X_0, X_1, \cdots, X_T)$, we obtain (7) immediately.

## B.2. Proof of Proposition 4.1

By the definition of SDE (1), we have

$$X_t|X_s \sim \mathcal{N}_d\left(m_{t|s} X_s, \sigma_{t|s}^2 I_d\right),$$

where $m_{t|s} = \exp\{-\int_s^t \beta_t \mathrm{d}t/2\}$ and $\sigma_{t|s}^2 = 1 - \exp\{-\int_s^t \beta_t \mathrm{d}t\}$. Then we have

$$
\begin{aligned}
\nabla_{X_t} \log q_t(X_t; \theta) &= \frac{1}{q_t(X_t; \theta)} \nabla_{X_t} q_t(X_t; \theta) \\
&= \frac{1}{q_t(X_t; \theta)} \nabla_{X_t} \int q_{t|s}(X_t|X_s) q_s(X_s; \theta) \mathrm{d}X_s \\
&= \frac{1}{q_t(X_t; \theta)} \int \nabla_{X_t} q_{t|s}(X_t|X_s) q_s(X_s; \theta) \mathrm{d}X_s \\
&= \int \frac{q_{t|s}(X_t|X_s) q_s(X_s; \theta)}{q_t(X_t; \theta)} \nabla_{X_t} \log q_{t|s}(X_t|X_s) \mathrm{d}X_s \\
&= \int q_{s|t}(X_s|X_t; \theta) \frac{m_{t|s} X_s - X_t}{\sigma_{t|s}^2} \mathrm{d}X_s \\
&= \frac{m_{t|s} \mathbb{E}(X_s|X_t) - X_t}{\sigma_{t|s}^2}
\end{aligned} \tag{19}
$$

which implies that

$$\mu_{s|t} = \mathbb{E}(X_s|X_t) = \frac{X_t + \sigma_{t|s}^2 \nabla_{X_t} \log q_t(X_t)}{m_{t|s}}. \tag{20}$$

For the covariance, note that

$$\Sigma_{s|t} = \mathbb{E}(X_s X_s^T | X_t) - \mathbb{E}(X_s|X_t) \mathbb{E}(X_s|X_t)^T.$$

To derive the covariance, we need to compute the second gradient of $\log q_t(X_t; \theta)$. By taking the gradient of (19) with respect to $X_t$ and using the same argument as above, we have

$$
\begin{aligned}
\nabla^2_{X_t} \log q_t(X_t; \theta) &= \int \frac{m_{t|s}X_s}{\sigma^2_{t|s}} \{\nabla_{X_t} q_{s|t}(X_s|X_t; \theta)\}^T dX_s - \frac{1}{\sigma^2_{t|s}} I_d \\
&= \int \frac{m_{t|s}X_s}{\sigma^2_{t|s}} q_{s|t}(X_s|X_t; \theta) \{\nabla_{X_t} \log q_{s|t}(X_s|X_t; \theta)\}^T dX_s - \frac{1}{\sigma^2_{t|s}} I_d \\
&= \int q_{s|t}(X_s|X_t; \theta) \frac{m_{t|s}X_s}{\sigma^2_{t|s}} \{\nabla_{X_t} \log q_{t|s}(X_t|X_s) - \nabla_{X_t} \log q_t(X_t; \theta)\}^T dX_s - \frac{1}{\sigma^2_{t|s}} I_d \\
&= \int q_{s|t}(X_s|X_t; \theta) \frac{m_{t|s}X_s}{\sigma^2_{t|s}} \{\nabla_{X_t} \log q_{t|s}(X_t|X_s)\}^T dX_s \\
&\quad - \frac{m_{t|s}\mathbb{E}(X_s|X_t)}{\sigma^2_{t|s}} \{\nabla_{X_t} \log q_t(X_t; \theta)\}^T - \frac{1}{\sigma^2_{t|s}} I_d \\
&= \int q_{s|t}(X_s|X_t; \theta) \frac{m_{t|s}X_s}{\sigma^2_{t|s}} \left\{ \frac{m_{t|s}X_s - X_t}{\sigma^2_{t|s}} \right\}^T dX_s \\
&\quad - \frac{m_{t|s}\mathbb{E}(X_s|X_t)}{\sigma^2_{t|s}} \left\{ \frac{m_{t|s}\mathbb{E}(X_s|X_t) - X_t}{\sigma^2_{t|s}} \right\}^T - \frac{1}{\sigma^2_{t|s}} I_d \\
&= \left( \frac{m_{t|s}}{\sigma^2_{t|s}} \right)^2 \{\mathbb{E}(X_s X_s^T|X_t) - \mathbb{E}(X_s|X_t)\mathbb{E}(X_s|X_t)^T\} - \frac{1}{\sigma^2_{t|s}} I_d.
\end{aligned}
$$

Hence, we conclude

$$
\Sigma_{s|t} = \frac{\sigma^4_{t|s}}{m^2_{t|s}} \nabla^2_{X_t} \log q_t(X_t; \theta) + \frac{\sigma^2_{t|s}}{m^2_{t|s}} I_d.
$$

This completes the proof.

### B.3. Proof of Theorem 5.5

From Pinsker's inequality, the first inequality is obvious. Thus, we focus on the second inequality. By data-processing inequality, we have

$$
\begin{aligned}
\mathrm{KL}(q_0||\tilde{p}_0) &\leq \mathrm{KL}(q_{0:T}||\tilde{p}_{0:T}) \\
&= \mathbb{E}_{X_{0:T} \sim q_{0:T}} \left[ \log \left( \frac{q_{0:T}(X_0, X_1, \cdots, X_T)}{\tilde{p}_{0:T}(X_0, X_1, \cdots, X_T)} \right) \right] \\
&= \mathbb{E}_{X_{0:T} \sim q_{0:T}} \left[ \log \left( \frac{q_T(X_T)}{\tilde{p}_T(X_T)} \right) + \sum_{t=1}^{T} \log \left( \frac{q_{t-1|t}(X_{t-1}|X_t)}{\tilde{p}_{t-1|t}(X_{t-1}|X_t)} \right) \right] \\
&= \underbrace{\mathrm{KL}(q_T||\tilde{p}_T)}_{\mathcal{I}_1: \text{ prior distribution error}} + \underbrace{\sum_{t=1}^{T} \mathbb{E}_{X_t \sim q_t} \left[ \mathrm{KL}(q_{t-1|t}(\cdot|X_t)||\tilde{p}_{t-1|t}(\cdot|X_t)) \right]}_{\mathcal{I}_2: \text{ transition density ratio error}}.
\end{aligned}
\tag{21}
$$

With the above decomposition, we now start to bound the two terms.

#### B.3.1. STEP 1: CONTROLLING THE PRIOR DISTRIBUTION ERROR

**Lemma B.1.** *Under Assumptions 5.1, we have*

$$
\mathrm{KL}(q_T||\tilde{p}_T) \leq \frac{1}{2}d\bar{\alpha}_T^2 + \frac{1}{2}\bar{\alpha}_T M_2 \lesssim \frac{d}{T^{2c_2}} + \frac{1}{T^{c_2}}
\tag{22}
$$

*for $T \geq 1$ and $c_2 \geq 1000$ is a large constant.*

The proof of Lemma B.1 can be found in Appendix B.5.1.

B.3.2. STEP 2: CONTROLLING THE TRANSITION DENSITY RATIO ERROR

We follow a similar argument in Li et al. (2024b) to bound the second term. To begin with, we define the following true posterior mean and covariance mapping:

$$\mu^*_{t-1|t}(X_t) = \frac{1}{\sqrt{\alpha_t}}(X_t + (1-\alpha_t)\nabla \log q_t(X_t)) := \frac{1}{\sqrt{\alpha_t}}(X_t + (1-\alpha_t)s^*_t(X_t)),$$

$$\Sigma^*_{t-1|t}(X_t) = \frac{1-\alpha_t}{\alpha_t}\{I_d + (1-\alpha_t)\nabla^2 \log q_t(X_t)\} := \frac{1-\alpha_t}{\alpha_t}\{I_d + (1-\alpha_t)H^*_t(X_t)\}.$$

(23)

and the estimated mapping as follows:

$$\hat{\mu}_{t-1|t}(X_t) = \frac{1}{\sqrt{\alpha_t}}(X_t + (1-\alpha_t)\hat{s}_t(X_t)),$$

$$\hat{\Sigma}_{t-1|t}(X_t) = \frac{1-\alpha_t}{\alpha_t}\{I_d + (1-\alpha_t)\hat{H}_t(X_t)\}.$$

(24)

It is clear that the transition density of $\tilde{Y}_{t-1}$ given $\tilde{Y}_t$ is

$$\tilde{p}_{t-1|t}(X_{t-1}|X_t) = \left(2\pi\frac{1-\alpha_t}{\alpha_t}\right)^{-\frac{d}{2}}\left|I_d + (1-\alpha_t)\hat{H}_t(X_t)\right|^{-\frac{1}{2}}$$

$$\cdot \exp\left\{-\frac{\alpha_t}{2(1-\alpha_t)}\left\|\left(I_d + (1-\alpha_t)\hat{H}_t(X_t)\right)^{-\frac{1}{2}}(X_{t-1} - \hat{\mu}_{t-1|t}(X_t))\right\|^2\right\}.$$

(25)

For any $t$, we introduce the following auxiliary sequences with the true score function and true Hessian function of the marginal density $q_t$ as follows:

$$H_{t-1} = \mu^*_{t-1|t}(H_t) + \Sigma^*_{t-1|t}(H_t)^{1/2}Z_t,$$

(26)

where $H_T \sim \mathcal{N}_d(0, I_d)$ and we define $p^H_t$ and $p^H_{t-1|t}$ as the marginal and transition density of $H_t$ and $H_{t-1}|H_t$. The transition density of $H_{t-1}$ given $H_t$ is given by

$$p^H_{t-1|t}(X_{t-1}|X_t) = \left(2\pi\frac{1-\alpha_t}{\alpha_t}\right)^{-\frac{d}{2}}\left|I_d + (1-\alpha_t)H^*_t(X_t)\right|^{-\frac{1}{2}}$$

$$\cdot \exp\left\{-\frac{\alpha_t}{2(1-\alpha_t)}\left\|(I_d + (1-\alpha_t)H^*_t(X_t))^{-\frac{1}{2}}(X_{t-1} - \mu^*_{t-1|t}(X_t))\right\|^2\right\}.$$

(27)

Hence, the term $\mathcal{I}_2$ can be bounded as follows:

$$\mathcal{I}_2 = \sum_{t=1}^{T}\mathbb{E}_{X_t\sim q_t}\left[\mathrm{KL}\left(q_{t-1|t}(\cdot|X_t)||\tilde{p}_{t-1|t}(\cdot|X_t)\right)\right]$$

$$= \underbrace{\sum_{t=1}^{T}\mathbb{E}_{X_t\sim q_t}\left\{\mathbb{E}_{X_{t-1}\sim q_{t-1|t}}\left[\log\frac{q_{t-1|t}(X_{t-1}|X_t)}{p^H_{t-1|t}(X_{t-1}|X_t)}\right]\right\}}_{\mathcal{I}_3: \text{ reverse step error}} + \underbrace{\sum_{t=1}^{T}\mathbb{E}_{X_t\sim q_t}\left\{\mathbb{E}_{X_{t-1}\sim q_{t-1|t}}\left[\log\frac{p^H_{t-1|t}(X_{t-1}|X_t)}{\tilde{p}_{t-1|t}(X_{t-1}|X_t)}\right]\right\}}_{\mathcal{I}_4: \text{ estimation error}}$$

(28)

To control the term $\mathcal{I}_3$, we introduce the following set in Li et al. (2024b) :

$$\mathcal{E} = \left\{(X_{t-1}, X_t) : -\log q_t(X_t) \lesssim d\log T, \|X_{t-1} - X_t/\sqrt{\alpha_t}\|^2 \lesssim \sqrt{d(1-\alpha_t)\log T}\right\}.$$

(29)

Turning to $q_{t-1|t}(X_{t-1}|X_t)$ over the set $\mathcal{E}$, we have the lemma as below:

**Lemma B.2.** *There exists some large enough numerical constant $c_s > 0$ such that: for any $(X_{t-1}, X_t) \in \mathcal{E}$, we have*

$$q_{t-1|t}(X_{t-1}|X_t)$$

$$= \left(2\pi\frac{1-\alpha_t}{\alpha_t}\right)^{-\frac{d}{2}}\left|I_d + (1-\alpha_t)H^*_t(X_t)\right|^{-\frac{1}{2}}$$

$$\cdot \exp\left\{-\frac{\alpha_t}{2(1-\alpha_t)}\left\|(I_d + (1-\alpha_t)H^*_t(X_t))^{-\frac{1}{2}}(X_{t-1} - \mu^*_{t-1|t}(X_t))\right\|^2 + \varepsilon_t(X_{t-1}, X_t)\right\},$$

(30)

*where the residual term $\varepsilon_t(X_{t-1}, X_t)$ satisfies*

$$|\varepsilon_t(X_{t-1}, X_t)| \leq c_s \frac{d^3 \log^{4.5} T}{T^{3/2}}. \tag{31}$$

The proof of Lemma B.2 is provided in Appendix B.5.2.

We can observe that under the set $\mathcal{E}$, the transition density $p^H_{t-1|t}(X_{t-1}|X_t)$ is nearly equal to the transition density $p_{H_{t-1}|H_t}(X_{t-1}|X_t)$ defined in Li et al. (2024a). With the proof of Lemma 11 in Li et al. (2024a) and using (53) and (54), we know that

$$(I_d + (1 - \alpha_t)H_t^*(X_t))^{-1} = (I_d + \frac{1}{2}(1 - \alpha_t)H_t^*(X_t))^{-2} + A, \tag{32}$$

where

$$\|A\| \lesssim \frac{d^2 \log^4 T}{T^2}. \tag{33}$$

Therefore, we have

$$\frac{p^H_{t-1|t}(X_{t-1}|X_t)}{p_{H_{t-1}|H_t}(X_{t-1}|X_t)} = 1 + O\left(\frac{d^3 \log^5 T}{T^2}\right). \tag{34}$$

Then, we introduce some useful lemmas established by Li et al. (2024a).

**Lemma B.3** (Lemma 11 in Li et al. (2024a)). *For every $(X_t, X_{t-1}) \in \mathcal{E}$, we have*

$$p_{H_{t-1}|H_t}(X_{t-1}|X_t)$$
$$\propto \exp\left\{-\frac{\alpha_t}{2(1-\alpha_t)}\left\|(I + (1-\alpha_t)H_t^*(X_t))^{-1}\left(X_{t-1} - \mu^*_{t-1|t}(X_t)\right)\right\|^2 + O\left(\frac{d^3 \log^5 T}{T^2}\right)\right\}. \tag{35}$$

**Lemma B.4** (Lemma 13 in Li et al. (2024a)). *For all $(X_t, X_{t-1}) \in \mathbb{R}^d \times \mathbb{R}^d$, we have*

$$\log \frac{q_{t-1|t}(X_{t-1}|X_t)}{p_{H_{t-1}|H_t}(X_{t-1}|X_t)} \leq T^{c_0+2c_R+2}\left\{\|X_{t-1} - X_t/\sqrt{\alpha_t}\|_2^2 + \|X_t\|_2^2 + 1\right\},$$

*where $c_0$ is defined in (16) and $c_R$ is defined in Lemma 3 in Li et al. (2024a).*

By (34), we know that Lemmas B.3 and B.4 can be applied in our cases. And with Lemma B.2, one can repeat the arguments in the proof of Lemma 14 in Li et al. (2024a), and get the same results as follows:

$$\mathcal{I}_3 = \sum_{t=1}^T \mathbb{E}_{X_t \sim q_t}\left[\mathrm{KL}\left(q_{t-1|t}(\cdot|X_t)\|p^H_{t-1|t}(\cdot|X_t)\right)\right] \lesssim \sum_{t=1}^T \frac{d^6 \log^9 T}{T^3} \asymp \frac{d^6 \log^9 T}{T^2}. \tag{36}$$

To control the term $\mathcal{I}_4$, we introduce the following lemma.

**Lemma B.5.** *Under Assumptions 5.2, 5.3 and 5.4, we have*

$$\sum_{t=1}^T \mathbb{E}_{X_t \sim q_t}\left\{\mathbb{E}_{X_{t-1} \sim q_{t-1|t}}\left[\log \frac{p^H_{t-1|t}(X_{t-1}|X_t)}{\tilde{p}_{t-1|t}(X_{t-1}|X_t)}\right]\right\} \lesssim \log T \varepsilon_s^2 + \frac{\log^2 T}{T}\varepsilon_H^2. \tag{37}$$

The proof of Lemma B.5 can be found in Appendix B.5.3.

Combining (36) and Lemma B.5 yields

$$\mathcal{I}_2 \lesssim \frac{d^6 \log^9 T}{T^2} + \log T \varepsilon_s^2 + \frac{\log^2 T}{T}\varepsilon_H^2. \tag{38}$$

Therefore, from Lemma B.1 and (38), we arrive at

$$\begin{aligned}
\mathrm{KL}(q_0\|\tilde{p}_0) &\lesssim \frac{d}{T^{2c_2}} + \frac{1}{T^{c_2}} + \frac{d^6 \log^9 T}{T^2} + \log T \varepsilon_s^2 + \frac{\log^2 T}{T}\varepsilon_H^2 \\
&\asymp \frac{d^6 \log^9 T}{T^2} + \log T \varepsilon_s^2 + \frac{\log^2 T}{T}\varepsilon_H^2
\end{aligned} \tag{39}$$

thereby concluding the proof of Theorem 5.5.

## B.4. Proof of Theorem 5.6

Denoted by

$$
\tilde{M}_{n,T}(\theta) := \frac{1}{n}\sum_{i=1}^{n}\log q_{0:T}(X_0^{(i)}, X_1^{(i)}, ..., X_T^{(i)}; \theta),
$$

$$
\hat{M}_{n,T}(\theta) := \frac{1}{n}\sum_{i=1}^{n}\sum_{t=1}^{T}\log \hat{p}_{t-1|t}(X_{t-1}^{(i)}|X_t^{(i)}; \theta) + \frac{1}{n}\sum_{i=1}^{n}\log \tilde{p}_T(X_T^{(i)}),
$$

$$
M_T(\theta) := \mathbb{E}_{X_{0:T}\sim q_{0:T}}\left[\log q_{0:T}(X_0, ..., X_T; \theta)\right],
$$

$$(40)$$

where $\tilde{p}_T(\cdot)$ denotes the density for $d$-dimensional standard normal distribution, $\hat{\theta}_{n,T} := \arg\min_\theta \mathcal{J}_{n,N}(\theta) = \arg\max_\theta \hat{M}_{n,T}(\theta)$, and $\tilde{\theta}_{n,T} := \arg\max_\theta \tilde{M}_{n,T}(\theta)$.

We assume that the following regularity conditions are satisfied

(1) The forward sampling procedure employs an equidistant grid with step size $\Delta$, which maintains an inverse proportionality relationship with the terminal time $T$.

(2) $\sup_\theta \left|\tilde{M}_{n,T}(\theta) - M_T(\theta)\right| \xrightarrow{p} 0$, as $n, T \to \infty$.

(3) For any $\epsilon > 0$, there exists a constant $\eta$, such that

$$
\sup_{|\theta-\theta^*|\geq\epsilon} M_T(\theta) < M_T(\theta^*) - \eta, \quad \text{for} \quad \forall n, T.
$$

(4) We suppose a uniform logarithmic approximation as follows:

$$
\sup_{\theta, x_0, \cdots, x_T}\left|\log\left(\frac{q_{0:T-1|T}(x_0, x_1, ..., x_{T-1}|x_T; \theta)}{\hat{p}_{0:T-1|T}(x_0, x_1, ..., x_{T-1}|x_T; \theta)}\right)\right| \leq \epsilon_2(T),
$$

where $\lim_{T\to\infty}\epsilon_2(T) = 0$.

The first two conditions are basically modified From Theorem 5.7 of Van der Vaart (2000) to ensure the consistency of true maximum likelihood estimation obtained from $\tilde{M}_{n,T}(\theta)$, i.e., $\tilde{\theta}_{n,T}$. And the third and fourth can be intuitively interpreted as the approximated likelihood behaves well, namely, the error can be uniformly bounded.

We observe that

$$
\begin{aligned}
\tilde{M}_{n,T}(\tilde{\theta}_{n,T}) &\geq \tilde{M}_{n,T}(\theta^*) \\
&= M_T(\theta^*) + \tilde{M}_{n,T}(\theta^*) - M_T(\theta^*) \\
&\geq M_T(\theta^*) - \sup_\theta\left|\tilde{M}_{n,T}(\theta) - M_T(\theta)\right|.
\end{aligned}
$$

$$(41)$$

Thus, combined with (41), we obtain

$$
\begin{aligned}
M_T(\theta^*) - M_T(\tilde{\theta}_{n,T}) &\leq \tilde{M}_{n,T}(\tilde{\theta}_{n,T}) - M_T(\tilde{\theta}_{n,T}) + \sup_\theta\left|\tilde{M}_{n,T}(\theta) - M_T(\theta)\right| \\
&\leq 2\sup_\theta\left|\tilde{M}_{n,T}(\theta) - M_T(\theta)\right|.
\end{aligned}
$$

$$(42)$$

Similarly to (41), we have the following results for $\hat{\theta}_{n,T}$, i.e.,

$$
\begin{aligned}
\hat{M}_{n,T}(\hat{\theta}_{n,T}) &\geq \hat{M}_{n,T}(\theta^*) \\
&= M_T(\theta^*) + \hat{M}_{n,T}(\theta^*) - M_T(\theta^*) \\
&\geq M_T(\theta^*) - \sup_\theta\left|\hat{M}_{n,T}(\theta) - M_T(\theta)\right|,
\end{aligned}
$$

$$(43)$$

and

$$
\begin{aligned}
M_T(\theta^*) - M_T(\hat{\theta}_{n,T}) &\leq \hat{M}_{n,T}(\hat{\theta}_{n,T}) - M_T(\hat{\theta}_{n,T}) + \sup_\theta \left| \hat{M}_{n,T}(\theta) - M_T(\theta) \right| \\
&\leq 2 \sup_\theta \left| \hat{M}_{n,T}(\theta) - M_T(\theta) \right|,
\end{aligned}
\tag{44}
$$

by plugging (43) into the first inequality.

Therefore, we are motivated to investigate how large $\sup_\theta \left| \hat{M}_{n,T}(\theta) - M_T(\theta) \right|$ will be. We notice that

$$
\sup_\theta \left| \hat{M}_{n,T}(\theta) - M_T(\theta) \right| \leq \sup_\theta \left| \tilde{M}_{n,T}(\theta) - M_T(\theta) \right| + \sup_\theta \left| \tilde{M}_{n,T}(\theta) - \hat{M}_{n,T}(\theta) \right|.
\tag{45}
$$

Since $\sup_\theta |\tilde{M}_{n,T}(\theta) - M_T(\theta)|$ is an $o_p(1)$ term, we only need to compute $\sup_\theta |\tilde{M}_{n,T}(\theta) - \hat{M}_{n,T}(\theta)|$. Notice that

$$
\begin{aligned}
&\tilde{M}_{n,T}(\theta) - \hat{M}_{n,T}(\theta) \\
&= \frac{1}{n} \sum_{i=1}^n \log \left( \frac{q_{0:T-1|T}(X_0^{(i)}, X_1^{(i)}, ..., X_{T-1}^{(i)}|X_T^{(i)};\theta)}{\hat{p}_{0:T-1|T}(X_0^{(i)}, X_1^{(i)}, ..., X_{T-1}^{(i)}|X_T^{(i)};\theta)} \right) + \frac{1}{n} \sum_{i=1}^n \log \left( \frac{q_T(X_T^{(i)})}{\tilde{p}_T(X_T^{(i)})} \right),
\end{aligned}
$$

and from Condition (3), we have

$$
\sup_{\theta, x_0, \cdots, x_T} \left| \log \left( \frac{q_{0:T-1|T}(x_0, x_1, ..., x_{T-1}|x_T;\theta)}{\hat{p}_{0:T-1|T}(x_0, x_1, ..., x_{T-1}|x_T;\theta)} \right) \right| \leq \epsilon_2(T).
\tag{46}
$$

Also, according to Lemma B.1 and the law of large numbers, we have

$$
\frac{1}{n} \sum_{i=1}^n \log \left( \frac{q_T(X_T^{(i)})}{\tilde{p}_T(X_T^{(i)})} \right) \xrightarrow{p} \mathrm{KL}(q_T || \tilde{p}_T),
\tag{47}
$$

which implies

$$
\frac{1}{n} \sum_{i=1}^n \log \left( \frac{q_T(X_T^{(i)})}{\tilde{p}_T(X_T^{(i)})} \right) = \epsilon_3(n,T),
$$

where $\epsilon_3(n,T) \xrightarrow{p} 0$, as $n$ and $T$ tend to $\infty$. Thus, we can decompose (44) as

$$
\begin{aligned}
M_T(\theta^*) - M_T(\hat{\theta}_{n,T}) &\leq 2 \left( \sup_\theta \left| \tilde{M}_{n,T}(\theta) - M_T(\theta) \right| + \sup_\theta \left| \tilde{M}_{n,T}(\theta) - \hat{M}_{n,T}(\theta) \right| \right) \\
&\leq 2 \sup_\theta \left| \tilde{M}_{n,T}(\theta) - M_T(\theta) \right| + 2\epsilon_2(T) + 2\epsilon_3(n,T).
\end{aligned}
$$

We observe that $\{\theta : |\theta - \theta^*| \geq \epsilon\} \subset \{\theta : M_T(\theta) < M_T(\theta^*) - \eta\}$. Thus, when $n, T$ is sufficiently large, such that $\sup_\theta |\tilde{M}_{n,T}(\theta) - M_T(\theta)| + \epsilon_2(T) + \epsilon_3(n,T) < \eta/2$, we obtain $|\hat{\theta}_{n,T} - \theta^*| < \epsilon$, which leads to

$$
\hat{\theta}_{n,T} \xrightarrow{p} \theta^*, \quad \text{as} \quad n, T \to \infty.
\tag{48}
$$

## B.5. Proof of auxiliary lemmas

### B.5.1. PROOF OF LEMMA B.1

Note that $\tilde{p}_T(X_T)$ is $\mathcal{N}_d(0, I_d)$ and $q_{t|0}(x|y) = \mathcal{N}(x; m_t y, \sigma_t^2 I_d)$, where $m_t = \sqrt{\bar{\alpha}_t}$ and $\sigma_t^2 = 1 - \bar{\alpha}_t$, we obtain

$$
\mathrm{KL}(q_{t|0}(\cdot|y) || \mathcal{N}_d(0, I_d)) = \frac{1}{2} \left( -d(1 - \sigma_t^2) - d \log \sigma_t^2 + m_t^2 \|y\|^2 \right).
$$

By the convexity of the KL divergence, we have

$$
\begin{aligned}
\mathrm{KL}(q_T||\mathcal{N}_d(0,I_d)) &= \mathrm{KL}\left(\int_{\mathbb{R}^d} q_{T|0}(x|y)\mathrm{d}Q_0(y)||\mathcal{N}_d(0,I_d)\right) \\
&\leq \int_{\mathbb{R}^d} \mathrm{KL}(q_{T|0}(\cdot|y)||\mathcal{N}_d(0,I_d))\mathrm{d}Q_0(y) \\
&= \frac{1}{2}\int_{\mathbb{R}^d}\left(-d(1-\sigma_T^2) - d\log\sigma_T^2 + m_T^2\|y\|^2\right)\mathrm{d}Q_0(y) \\
&= \frac{1}{2}\left(-d(1-\sigma_T^2) - d\log\sigma_T^2 + m_T^2\mathbb{E}_{X\sim q_0}\|X\|^2\right) \\
&\leq \frac{1}{2}(-d\bar{\alpha}_T - d\log(1-\bar{\alpha}_T) + \bar{\alpha}_T M_2).
\end{aligned}
\tag{49}
$$

Since $\log(1+x) > x - x^2$ when $x > -0.68$ and $\bar{\alpha}_T < 0.68$ when $T \geq 1$, we obtain

$$
-\log(1-\bar{\alpha}_T) < \bar{\alpha}_T + \bar{\alpha}_T^2.
$$

Thus

$$
\mathrm{KL}(q_T||\tilde{p}_T) \leq \frac{1}{2}d\bar{\alpha}_T^2 + \frac{1}{2}\bar{\alpha}_T M_2 \lesssim \frac{d}{T^{2c_2}} + \frac{1}{T^{c_2}},
\tag{50}
$$

where $c_2 \geq 1000$ and the last inequality holds by the properties of the noise schedule in Li et al. (2024b).

### B.5.2. PROOF OF LEMMA B.2

Lemma 12 in Li et al. (2024b) shows that the transition density of $X_{t-1}$ given $X_t$ can be expressed as

$$
q_{t-1|t}(X_{t-1}|X_t) = f_1(X_t)\exp\left\{-f_2(X_{t-1},X_t) + \varepsilon_{t,1}(X_{t-1},X_t)\right\}
\tag{51}
$$

for some function $f_1(\cdot)$, where

$$
\begin{aligned}
&f_2(X_{t-1},X_t) \\
&= \frac{\alpha_t}{2(1-\alpha_t)}\left\{\left(X_{t-1}-\mu_{t-1|t}^*(X_t)\right)^T(I_d - (1-\alpha_t)H_t^*(X_t))\left(X_{t-1}-\mu_{t-1|t}^*(X_t)\right)\right\}
\end{aligned}
\tag{52}
$$

and

$$
|\varepsilon_{t,1}(X_{t-1},X_t)| \lesssim \frac{d^3\log^{4.5}T}{T^{3/2}}.
$$

Note that the formulation of the covariance matrix $I_d - (1-\alpha_t)H_t^*(X_t)$ still differs from $(I_d + (1-\alpha_t)\hat{H}_t(X_t))^{-1}$. Following the same procedure in Li et al. (2024b), we can show that

$$
(I_d + (1-\alpha_t)H_t^*(X_t))^{-1} = I_d - (1-\alpha_t)H_t^*(X_t) + A,
\tag{53}
$$

where $A$ is a matrix obeying

$$
\|A\| \lesssim \frac{d^2\log^4 T}{T^2}.
\tag{54}
$$

Combining the above, we arrive at

$$
q_{t-1|t}(X_{t-1}|X_t) = f_3(X_t)\exp\left\{-f_4(X_{t-1},X_t) + \varepsilon_{t,2}(X_{t-1},X_t)\right\}
\tag{55}
$$

for some function $f_3(\cdot)$, where

$$
\begin{aligned}
&f_4(X_{t-1},X_t) \\
&= \frac{\alpha_t}{2(1-\alpha_t)}\left\{\left(X_{t-1}-\mu_{t-1|t}^*(X_t)\right)^T(I_d - (1-\alpha_t)H_t^*(X_t))^{-1}\left(X_{t-1}-\mu_{t-1|t}^*(X_t)\right)\right\}
\end{aligned}
\tag{56}
$$

and

$$
|\varepsilon_{t,2}(X_{t-1},X_t)| \lesssim \frac{d^3\log^{4.5}T}{T^{3/2}}.
$$

Repeating Step 3 in the proof of Lemma 8 in Li et al. (2024b), it yields that

$$f_3(X_t) = \left(1 + O\left(\frac{d^3 \log^{4.5} T}{T^{3/2}}\right)\right)\left(2\pi \frac{1 - \alpha_t}{\alpha_t}\right)^{-\frac{d}{2}} |I_d + (1 - \alpha_t)H_t^*(X_t)|^{-\frac{1}{2}}.$$

This completes the proof.

### B.5.3. PROOF OF LEMMA B.5

Considering the approach in Liang et al. (2025), we directly calculate the density ratio between two Gaussian distributions with the different mean and different covariance. We have

$$
\begin{aligned}
&\log \frac{p_{t-1|t}^H(X_{t-1}|X_t)}{\tilde{p}_{t-1|t}(X_{t-1}|X_t)} \\
=& \frac{1}{2}\log\left(\frac{\det\left(I_d + (1 - \alpha_t)\hat{H}_t(X_t)\right)}{\det(I_d + (1 - \alpha_t)H_t^*(X_t))}\right) + \frac{\alpha_t}{2(1 - \alpha_t)}(X_{t-1} - \hat{\mu}_{t-1|t}(X_t))^T \\
&\quad \cdot \left\{(I_d + (1 - \alpha_t)\hat{H}_t(X_t))^{-1} - (I_d + (1 - \alpha_t)H_t^*(X_t))^{-1}\right\}(X_{t-1} - \hat{\mu}_{t-1|t}(X_t)) \\
&+ \frac{\alpha_t}{2(1 - \alpha_t)}(X_{t-1} - \hat{\mu}_{t-1|t}(X_t))^T(I_d + (1 - \alpha_t)H_t^*(X_t))^{-1}(X_{t-1} - \hat{\mu}_{t-1|t}(X_t)) \\
&- \frac{\alpha_t}{2(1 - \alpha_t)}(X_{t-1} - \mu_{t-1|t}^*(X_t))^T(I_d + (1 - \alpha_t)H_t^*(X_t))^{-1}(X_{t-1} - \mu_{t-1|t}^*(X_t)) \\
=& \frac{1}{2}\log\left(\frac{\det\left(I_d + (1 - \alpha_t)\hat{H}_t(X_t)\right)}{\det(I_d + (1 - \alpha_t)H_t^*(X_t))}\right) + \frac{\alpha_t}{2(1 - \alpha_t)}(\mu_{t-1|t}^*(X_t) - \hat{\mu}_{t-1|t}(X_t))^T \\
&\quad \cdot (I_d + (1 - \alpha_t)H_t^*(X_t))^{-1}(\mu_{t-1|t}^*(X_t) - \hat{\mu}_{t-1|t}(X_t)) \\
&+ \frac{\alpha_t}{2(1 - \alpha_t)}(X_{t-1} - \mu_{t-1|t}^*(X_t))^T\left\{(I_d + (1 - \alpha_t)\hat{H}_t(X_t))^{-1} - (I_d + (1 - \alpha_t)H_t^*(X_t))^{-1}\right\} \\
&\quad \cdot (X_{t-1} - \mu_{t-1|t}^*(X_t)) \\
&+ \frac{\alpha_t}{2(1 - \alpha_t)}(X_{t-1} - \mu_{t-1|t}^*(X_t))^T\left\{(I_d + (1 - \alpha_t)\hat{H}_t(X_t))^{-1} - (I_d + (1 - \alpha_t)H_t^*(X_t))^{-1}\right\} \\
&\quad \cdot (\mu_{t-1|t}^*(X_t) - \hat{\mu}_{t-1|t}(X_t)) \\
&+ \frac{\alpha_t}{2(1 - \alpha_t)}(\mu_{t-1|t}^*(X_t) - \hat{\mu}_{t-1|t}(X_t))^T\left\{(I_d + (1 - \alpha_t)\hat{H}_t(X_t))^{-1}\right. \\
&\quad \left. - (I_d + (1 - \alpha_t)H_t^*(X_t))^{-1}\right\} \cdot (X_{t-1} - \mu_{t-1|t}^*(X_t)).
\end{aligned}
\tag{57}
$$

For the last two terms in (57), we can observe that under the expectation w.r.t $X_{t-1} \sim q_{t-1|t}$, they are both zero. Thus, by a little algebra, we have

$$
\begin{aligned}
&\mathbb{E}_{X_{t-1}\sim q_{t-1|t}} \log \frac{p_{t-1|t}^H(X_{t-1}|X_t)}{\tilde{p}_{t-1|t}(X_{t-1}|X_t)} \\
=& \frac{1}{2}\log\left(\frac{\det\left(I_d + (1 - \alpha_t)\hat{H}_t(X_t)\right)}{\det(I_d + (1 - \alpha_t)H_t^*(X_t))}\right) \\
&+ \frac{\alpha_t}{2(1 - \alpha_t)}(\mu_{t-1|t}^*(X_t) - \hat{\mu}_{t-1|t}(X_t))^T(I_d + (1 - \alpha_t)H_t^*(X_t))^{-1}(\mu_{t-1|t}^*(X_t) - \hat{\mu}_{t-1|t}(X_t)) \\
&+ \frac{1}{2}\mathbb{E}_{X_{t-1}\sim q_{t-1|t}} \text{tr}\left[(I_d + (1 - \alpha_t)\hat{H}_t(X_t))^{-1}(I_d + (1 - \alpha_t)H_t^*(X_t)) - d\right].
\end{aligned}
\tag{58}
$$

Considering the second term in (58) and from Assumption 5.4, we obtain that

$$\frac{\alpha_t}{2(1-\alpha_t)}\mathbb{E}_{X_t\sim q_t}\left[(\mu^*_{t-1|t}(X_t)-\hat{\mu}_{t-1|t}(X_t))^T(I_d+(1-\alpha_t)H^*_t(X_t))^{-1}\cdot(\mu^*_{t-1|t}(X_t)-\hat{\mu}_{t-1|t}(X_t))\right]$$

$$=\frac{1-\alpha_t}{2}\mathbb{E}_{X_t\sim q_t}\left[(s^*_t(X_t)-\hat{s}_t(X_t))^T(I_d+(1-\alpha_t)H^*_t(X_t))^{-1}(s^*_t(X_t)-\hat{s}_t(X_t))\right]$$

$$\leq\frac{1-\alpha_t}{2}\mathbb{E}_{X_t\sim q_t}\left[\|s^*_t(X_t)-\hat{s}_t(X_t)\|^2\|(I_d+(1-\alpha_t)H^*_t(X_t))^{-1}\|\right] \tag{59}$$

$$\leq\frac{1-\alpha_t}{2(1+\varepsilon_0)}\mathbb{E}_{X_t\sim q_t}\|s^*_t(X_t)-\hat{s}_t(X_t)\|^2$$

$$\lesssim\frac{1-\alpha_t}{2}\mathbb{E}_{X_t\sim q_t}\|s^*_t(X_t)-\hat{s}_t(X_t)\|^2.$$

For the first term in (58), the term $1-\alpha_t$ is small enough when $t$ is large, thus we can use Taylor expansion to show that

$$\mathbb{E}_{X_t\sim q_t}\log(\det(I_d+(1-\alpha_t)H^*_t(X_t)))$$

$$=\mathbb{E}_{X_t\sim q_t}\log\left(1+(1-\alpha_t)\operatorname{tr}(H^*_t(X_t))+\frac{(1-\alpha_t)^2}{2}\operatorname{tr}(H^*_t(X_t))^2-\frac{(1-\alpha_t)^2}{2}\operatorname{tr}(H^*_t(X_t)^2)+O((1-\alpha_t)^3)\right) \tag{60}$$

$$=\mathbb{E}_{X_t\sim q_t}\left[(1-\alpha_t)\operatorname{tr}(H^*_t(X_t))-\frac{(1-\alpha_t)^2}{2}\operatorname{tr}(H^*_t(X_t)^2)\right]+O((1-\alpha_t)^3).$$

Thus, by the same argument, we get

$$\frac{1}{2}\mathbb{E}_{X_t\sim q_t}\log\left(\frac{\det\left(I_d+(1-\alpha_t)\hat{H}_t(X_t)\right)}{\det(I_d+(1-\alpha_t)H^*_t(X_t))}\right)$$

$$=\frac{1}{2}\mathbb{E}_{X_t\sim q_t}\left[(1-\alpha_t)\operatorname{tr}\left(\hat{H}_t(X_t)-H^*_t(X_t)\right)-\frac{(1-\alpha_t)^2}{2}\operatorname{tr}\left(\hat{H}_t(X_t)^2-H^*_t(X_t)^2\right)\right]+O((1-\alpha_t)^3). \tag{61}$$

For the third term in (58), we have

$$\frac{1}{2}\mathbb{E}_{X_{t-1}\sim q_{t-1|t}}\operatorname{tr}\left[(I_d+(1-\alpha_t)\hat{H}_t(X_t))^{-1}(I_d+(1-\alpha_t)H^*_t(X_t))-d\right]$$

$$=\frac{1}{2}\mathbb{E}_{X_{t-1}\sim q_{t-1|t}}\operatorname{tr}\left[(I_d-(1-\alpha_t)\hat{H}_t(X_t)+(1-\alpha_t)^2\hat{H}_t(X_t)^2+O((1-\alpha_t)^3))(I_d+(1-\alpha_t)H^*_t(X_t))-d\right] \tag{62}$$

$$=\frac{1}{2}\mathbb{E}_{X_{t-1}\sim q_{t-1|t}}\operatorname{tr}\left[(1-\alpha_t)\operatorname{tr}(H^*_t(X_t))-(1-\alpha_t)\operatorname{tr}\left(\hat{H}_t(X_t)\right)+(1-\alpha_t)^2\operatorname{tr}\left(\hat{H}_t(X_t)^2\right)\right]+O((1-\alpha_t)^2).$$

Combining (59), (61) and (62), we arrive at

$$\mathbb{E}_{X_t\sim q_t}\left\{\mathbb{E}_{X_{t-1}\sim q_{t-1|t}}\log\frac{p^H_{t-1|t}(X_{t-1}|X_t)}{\tilde{p}_{t-1|t}(X_{t-1}|X_t)}\right\}$$

$$\lesssim\frac{(1-\alpha_t)^2}{2\alpha_t}\mathbb{E}_{X_t\sim q_t}\|s^*_t(X_t)-\hat{s}_t(X_t)\|^2$$

$$+\frac{(1-\alpha_t)^2}{2}\mathbb{E}_{X_t\sim q_t}\left[\operatorname{tr}\left(\hat{H}_t(X_t)^2\right)+\operatorname{tr}(H^*_t(X_t)^2)-2\operatorname{tr}\left(\hat{H}_t(X_t)H^*_t(X_t)\right)\right]+O((1-\alpha_t)^2) \tag{63}$$

$$=\frac{1-\alpha_t}{2}\mathbb{E}_{X_t\sim q_t}\|s^*_t(X_t)-\hat{s}_t(X_t)\|^2+\frac{(1-\alpha_t)^2}{2}\mathbb{E}_{X_t\sim q_t}\operatorname{tr}\left((\hat{H}_t(X_t)-H^*_t(X_t))^2\right)+O((1-\alpha_t)^2)$$

$$=\frac{1-\alpha_t}{2}\mathbb{E}_{X_t\sim q_t}\|s^*_t(X_t)-\hat{s}_t(X_t)\|^2+\frac{(1-\alpha_t)^2}{2}\mathbb{E}_{X_t\sim q_t}\left\|\hat{H}_t(X_t)-H^*_t(X_t)\right\|^2_F+O((1-\alpha_t)^2).$$

Consequently, we can demonstrate that

$$\sum_{t=1}^T\mathbb{E}_{X_t\sim q_t}\left\{\mathbb{E}_{X_{t-1}\sim q_{t-1|t}}\log\frac{p^H_{t-1|t}(X_{t-1}|X_t)}{\tilde{p}_{t-1|t}(X_{t-1}|X_t)}\right\}\lesssim\frac{1-\alpha_t}{2}T\varepsilon_s^2+\frac{(1-\alpha_t)^2}{2}T\varepsilon_H^2$$

$$\lesssim\log T\varepsilon_s^2+\frac{\log^2 T}{T}\varepsilon_H^2. \tag{64}$$

This completes the proof.

# C. Experiment Details

## C.1. Empirical Conditioning Diagnostics

To provide empirical evidence related to Assumption 5.4, we report the minimum eigenvalue of the sample covariance of forward-noised CIFAR-10 at different noise levels Table 2.

*Table 2.* Minimum eigenvalue of the sample covariance of forward-noised CIFAR-10 at different noise levels $t/T$. Even mild Gaussian smoothing moves the empirical distribution away from degeneracy.

| $t/T$ | 0.001 | 0.005 | 0.010 | 0.020 | 0.050 | 0.100 | 0.500 | 0.750 |
|---|---|---|---|---|---|---|---|---|
| $\lambda_{\min}$ | $3.25 \times 10^{-4}$ | $1.39 \times 10^{-3}$ | $2.58 \times 10^{-3}$ | $4.87 \times 10^{-3}$ | $1.15 \times 10^{-2}$ | $2.22 \times 10^{-2}$ | $1.02 \times 10^{-1}$ | $1.51 \times 10^{-1}$ |

## C.2. Experiment Setting

**Synthetic 1D and 2D mixture experiments.** We conducted experiments on synthetic 1D and 2D mixture distributions to evaluate the performance of our Likelihood Matching (LM) and Score Matching (SM) methods under controlled conditions. In the non-oracle setting, where the true parametric form of the data distribution is unknown, we trained fully connected neural networks with a single hidden layer and ReLU activation functions to approximate the score and covariance terms. Models were trained for 500 epochs using the Adam optimizer with a learning rate of 0.01 and full-batch gradient descent.

**Real image datasets.** We further evaluated our method on several standard image generation benchmarks: MNIST ($32 \times 32$ grayscale, Deng 2012), CIFAR-10 ($32 \times 32$ RGB), CelebA ($64 \times 64$ RGB, Liu et al. 2015), LSUN Church and LSUN Bedroom ($64 \times 64$ RGB, Yu et al. 2015). All image data were normalized to the range $[-1, 1]$.

We adopted a U-Net architecture for both the score function and the Hessian function approximation, following previous work in score-based diffusion modeling. For the Hessian network, the number of output channels is set to $(r+1) \times C$, where $r$ is the predefined low-rank parameter and $C$ denotes the number of image channels. The Hessian function is modeled using a spiked structure following Meng et al. (2021):

$$H_t(X_t; \phi) = \boldsymbol{U}_t(X_t; \phi) + \boldsymbol{V}_t(X_t; \phi)\boldsymbol{V}_t(X_t; \phi)^T,$$

where $\boldsymbol{U}_t \in \mathbb{R}^d$ is a diagonal matrix and $\boldsymbol{V}_t \in \mathbb{R}^{d \times r}$ represents the low-rank component. We applied a ReLU activation to the output of $\boldsymbol{U}_t$ to ensure the positive definiteness of $H_t$.

In the experiments, we set time steps $T = 1$. For clarity, the score network uses a standard U-Net with 4 down/up blocks, 2 ResNet layers per block, and channels (128, 256, 256, 512), with attention in the third down and second up blocks. The Hessian network follows the same structure but uses 1 ResNet layer per block and smaller channels (64, 128, 128, 128). Its output is $(1 + r)$ times the input channels, representing a diagonal-plus-low-rank structure following Meng et al. (2021).

All models were trained for 500,000 iterations using the Adam optimizer with $(\beta_1, \beta_2) = (0.9, 0.999)$ and a learning rate of $10^{-4}$. We adopted a linear noise schedule with $\beta(0) = 0.1$ and $\beta(T) = 20$, consistent with the settings in Song et al. (2021c). Training was performed on NVIDIA A100 GPUs. The batch size was set to 128 for MNIST and CIFAR-10, and 64 for CelebA, LSUN Church, and LSUN Bedroom. We applied Exponential Moving Average (EMA) to model parameters with a decay rate of 0.9999 to improve stability during training and sampling. For evaluation, we computed the Fréchet Inception Distance (FID) using the `torchmetrics` module with feature dimension 2048. FID was calculated based on 10,000 generated samples per dataset. Prior to evaluation, all images were resized and center-cropped to $299 \times 299$ pixels, and grayscale images (e.g., MNIST) were replicated across the RGB channels to match the input format of the InceptionV3 model. For likelihood evaluation, we compute the NLL directly under the discrete SDE by evaluating the exact Gaussian likelihood of the residuals using the learned covariance.

**Compatibility with conditional generation.** LM modifies the training objective but keeps the score-network interface unchanged. Therefore, sampling-time techniques that only require score predictions remain applicable. Classifier-free guidance can be applied by interpolating conditional and unconditional score predictions; LM-trained scores can also serve as teachers for consistency distillation; and DDIM or ODE-based samplers can use the LM-trained score without using the Hessian network. The Hessian is required only for our Hessian-informed stochastic sampler.

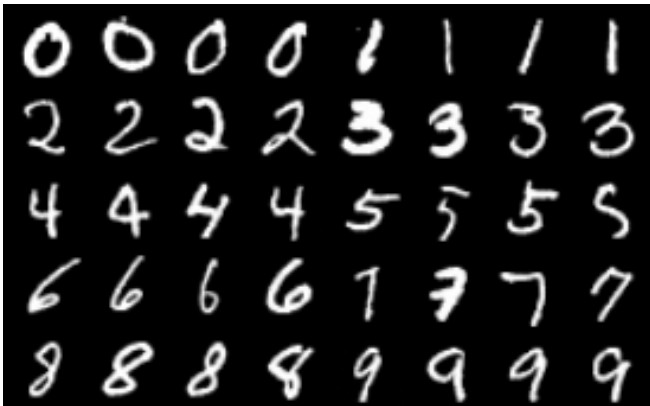

*Figure 5.* Class-conditional MNIST samples generated by LM. Prompt IDs corresponding to digits zero through nine are fed into the same conditional U-Net used by the score and covariance models. The samples are prompt-consistent, showing that LM is compatible with conditional generation pipelines.

### C.3. Additional Results

Figure 7 shows us the comparisons between the discrepancies between the original data (Mixture Gaussian) and the synthetic data by LM. In particular, for the one-dimensional case, we use the kernel density estimation tool to visualize the densities obtained from both synthetic data and the original data in Figure 7a. For the two-dimensional case, we illustrate the difference between the synthetic data and the original data using scatter plots in Figure 7b. From both figures, we see that our synthetic data indeed learn the associated underlying densities of the original data.

Since the ground-truth density is available in the 1D Gaussian mixture, we also evaluate held-out NLL and $\text{KL}(q_0\|\hat{p}_0)$ using KDE-based density estimates. Table 3 showed that LM improves over SM not only under MMD but also under likelihood-oriented metrics.

*Table 3.* Likelihood-oriented evaluation on the 1D Gaussian mixture. Lower is better for all metrics. The learned density is estimated by KDE for both LM and SM.

| Metric | True/Oracle | LM | SM |
|---|---|---|---|
| Held-out NLL | 2.144 | 2.192 | 2.367 |
| $\text{KL}(q_0\|\hat{p}_0)$ | – | 0.251 | 0.264 |
| MMD | – | $5.32\times10^{-4}$ | $2.36\times10^{-3}$ |

In all experiments, we did not observe divergence when training the Hessian model (Figure 6). Stability is aided by three design choices: the covariance is kept positive definite through the diagonal-plus-low-rank parameterization, the rank constraint regularizes the second-order model, and EMA with decay 0.9999 smooths parameter updates.

Figure 9 also presents sample generations from the Likelihood Matching method ($N = 2, r = 10$) on the CIFAR10, CelebA, LSUN Church and LSUN Bedroom datasets.

### C.4. Computational Analysis

While introducing a Hessian network increases computational overhead, our framework remains scalable due to the low-rank approximation and efficient implementation using the Sherman-Morrison-Woodbury formula (see Appendix C.5). On a single A100 GPU for CIFAR-10, training time per iteration increased from 0.291s (SM) to a manageable 0.599s for a diagonal Hessian ($r = 0$) and 0.756s for $r = 200$. Similarly, sampling time per 1000 steps grew from 12.66s to 27.65s. This analysis demonstrates a favorable trade-off between performance gains and computational cost. A detailed breakdown of runtime and memory usage is available in Table 5.

Memory usage also scaled controllably with Hessian rank $r$, remaining well within the practical limits. Specifically, the full training required 36.2GB to 40.4GB ($r = 0$ to 200), representing 1.7-2.1 times of the SM baseline (17.2GB). Crucially,

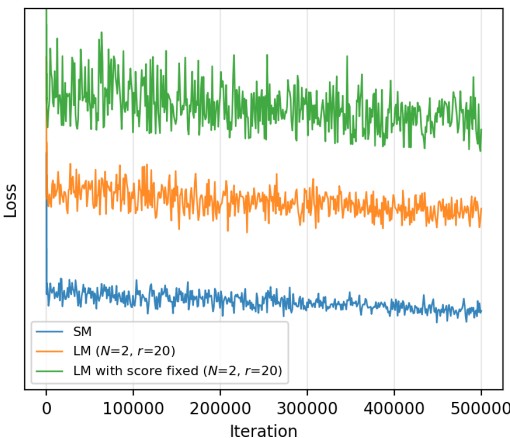

*Figure 6.* Training stability curves on MNIST. Since SM and LM optimize different objectives, the absolute loss values are not directly comparable; the figure is intended to show that LM training remains stable without divergence. The diagonal-plus-low-rank Hessian parameterization, positive covariance construction, and EMA smoothing help prevent instability from high-dimensional second-order information.

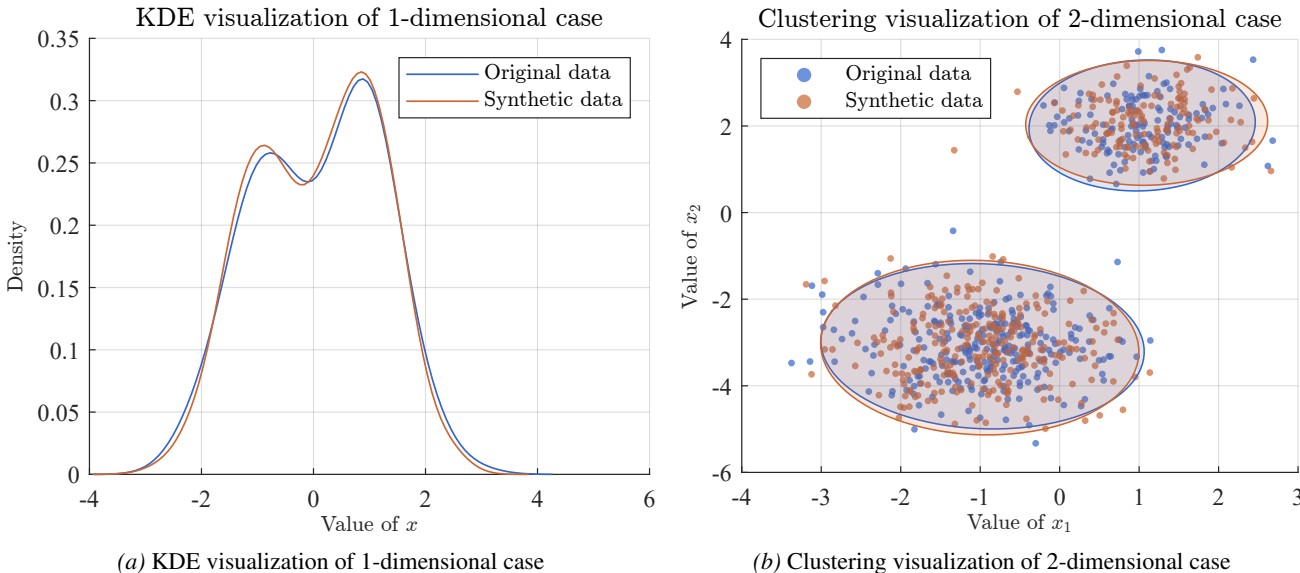

(a) KDE visualization of 1-dimensional case   (b) Clustering visualization of 2-dimensional case

*Figure 7.* Comparison of original and synthetic data. (a) Kernel Density Estimations (KDE) for the 1-dimensional case. (b) Clustering results for the 2-dimensional case.

*Table 4.* Comparison of parameter estimations for the two-dimensional mixture

(a) Estimation by Likelihood Matching

| Parameter | $n = 100$ | | $n = 200$ | |
|---|---|---|---|---|
| | MAE | Std. Error | MAE | Std. Error |
| $\mu_{11}$ | **0.0840** | **0.1065** | **0.0596** | **0.0733** |
| $\mu_{12}$ | **0.0838** | **0.1045** | **0.0549** | **0.0698** |
| $\mu_{21}$ | **0.0842** | **0.1057** | **0.0591** | **0.0727** |
| $\mu_{22}$ | **0.0841** | **0.1039** | **0.0627** | **0.0783** |
| $\sigma_1$ | 0.2550 | **0.0556** | 0.2520 | **0.0364** |
| $\sigma_2$ | 0.1831 | **0.0753** | **0.1814** | **0.0531** |
| $\omega_1$ | **0.1249** | **0.1529** | **0.0829** | **0.1042** |

(b) Estimation by Score Matching

| Parameter | $n = 100$ | | $n = 200$ | |
|---|---|---|---|---|
| | MAE | Std. Error | MAE | Std. Error |
| $\mu_{11}$ | 0.1137 | 0.1408 | 0.0800 | 0.0985 |
| $\mu_{12}$ | 0.1092 | 0.1344 | 0.0768 | 0.0955 |
| $\mu_{21}$ | 0.1185 | 0.1500 | 0.0853 | 0.1079 |
| $\mu_{22}$ | 0.1164 | 0.1480 | 0.0840 | 0.1064 |
| $\sigma_1$ | **0.2519** | 0.0745 | **0.2468** | 0.0508 |
| $\sigma_2$ | **0.1818** | 0.0990 | 0.1820 | 0.0707 |
| $\omega_1$ | 0.1566 | 0.1923 | 0.1154 | 0.1409 |

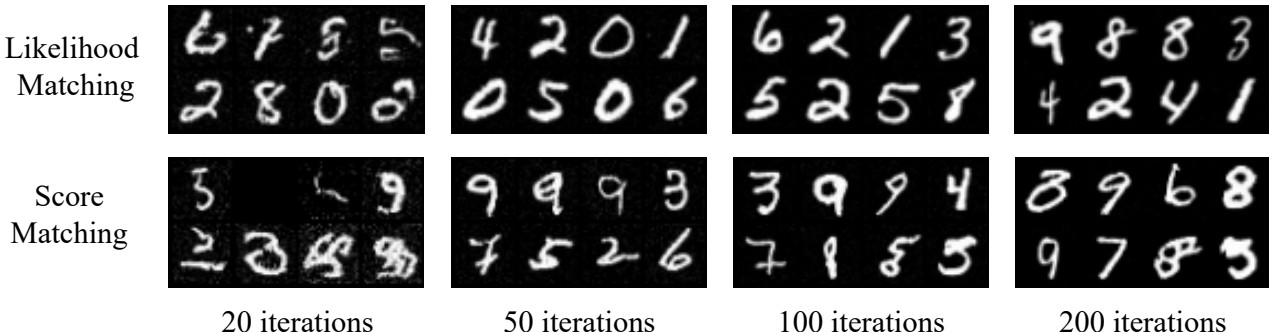

| Likelihood Matching | | | | |
| Score Matching | | | | |
| 20 iterations | 50 iterations | 100 iterations | 200 iterations |

*Figure 8.* Sampling on MNIST. Both Likelihood Matching and Score Matching use the sampler (12), with the Hessian function set to zero in the case of Score Matching.

our Hessian-only training mode, where the score network remained fixed, reduced the overhead to just 8.3GB ($r = 0$) to 13.3GB ($r = 200$). Growing $r$ from 100 to 200 increased memory by only 16% (11.5GB to 13.3GB), demonstrating efficient memory management even at high approximation fidelity.

The computing comparison results on high-resolution ImageNet are shown in Table 6, which indicate that the computational burden increases notably at higher resolution and may require further, dedicated research to fully address.

*Table 5.* Training and sampling cost of the LM with different Hessian ranks $r$ on CIFAR-10 (A100, batch size 256). "Hessian Time" and "Hessian Mem" refer to the additional cost of training the Hessian alone with a fixed score network.

| | Training Time (s/it) | Training Mem (MB) | Hessian Time (s/it) | Hessian Mem (MB) | Sampling Time (s/1000 iters) |
|---|---|---|---|---|---|
| SM | 0.291 | 17,247 | / | / | 12.66 |
| LM ($r = 0$) | 0.599 | 36,220 | 0.303 | 8,286 | 20.83 |
| LM ($r = 20$) | 0.617 | 36,428 | 0.324 | 8,418 | 21.48 |
| LM ($r = 100$) | 0.664 | 38,822 | 0.369 | 11,452 | 23.61 |
| LM ($r = 200$) | 0.756 | 40,444 | 0.463 | 13,302 | 27.65 |

*Table 6.* Training and sampling cost of the LM with different Hessian ranks $r$ on 224×224 ImageNet (A100, batch size 4). "Hessian Time" and "Hessian Mem" refer to the additional cost of training the Hessian alone with a fixed score network.

| | Training Time (s/it) | Training Mem (MB) | Hessian Time (s/it) | Hessian Mem (MB) | Sampling Time (s/1000 iters) |
|---|---|---|---|---|---|
| SM | 0.155 | 17,183 | / | / | 34.6 |
| LM ($r = 0$) | 0.566 | 49,287 | 0.362 | 28,943 | 68.5 |
| LM ($r = 20$) | 0.571 | 49,641 | 0.370 | 29,057 | 69.8 |
| LM ($r = 100$) | 0.583 | 50,213 | 0.384 | 29,913 | 71.2 |
| LM ($r = 200$) | 0.598 | 51,357 | 0.403 | 31,163 | 74.0 |

### C.5. Efficient Implementation of Training and Sampling Procedure

Likelihood Matching training and inference involve repeated evaluations of computationally intensive linear algebra operations, including matrix inversion, matrix square roots, and determinant calculations. Given that image data typically resides in high-dimensional spaces (e.g., $d > 1000$), the associated computational cost, on the order of $\mathcal{O}(d^3)$, becomes prohibitive in practice. To mitigate this issue, we adopt the diagonal-plus-low-rank covariance parameterization proposed by Meng et al. (2021), modeling the covariance as

$$H_t(X_t; \phi) = \boldsymbol{U}_t(X_t; \phi) + \boldsymbol{V}_t(X_t; \phi)\boldsymbol{V}_t(X_t; \phi)^T,$$

where $\boldsymbol{U}_t(\cdot; \phi) : \mathbb{R}^d \to \mathbb{R}^{d \times d}$ is a diagonal matrix, and $\boldsymbol{V}_t(\cdot; \phi) : \mathbb{R}^d \to \mathbb{R}^{d \times r}$ is a low-rank matrix with a prespecified rank $r \ll d$. For notational simplicity, we omit the dependence on $(X_t; \phi)$ and associated superscripts/subscripts.

This structural assumption enables a series of simplifications that substantially reduce the computational cost of matrix operations.

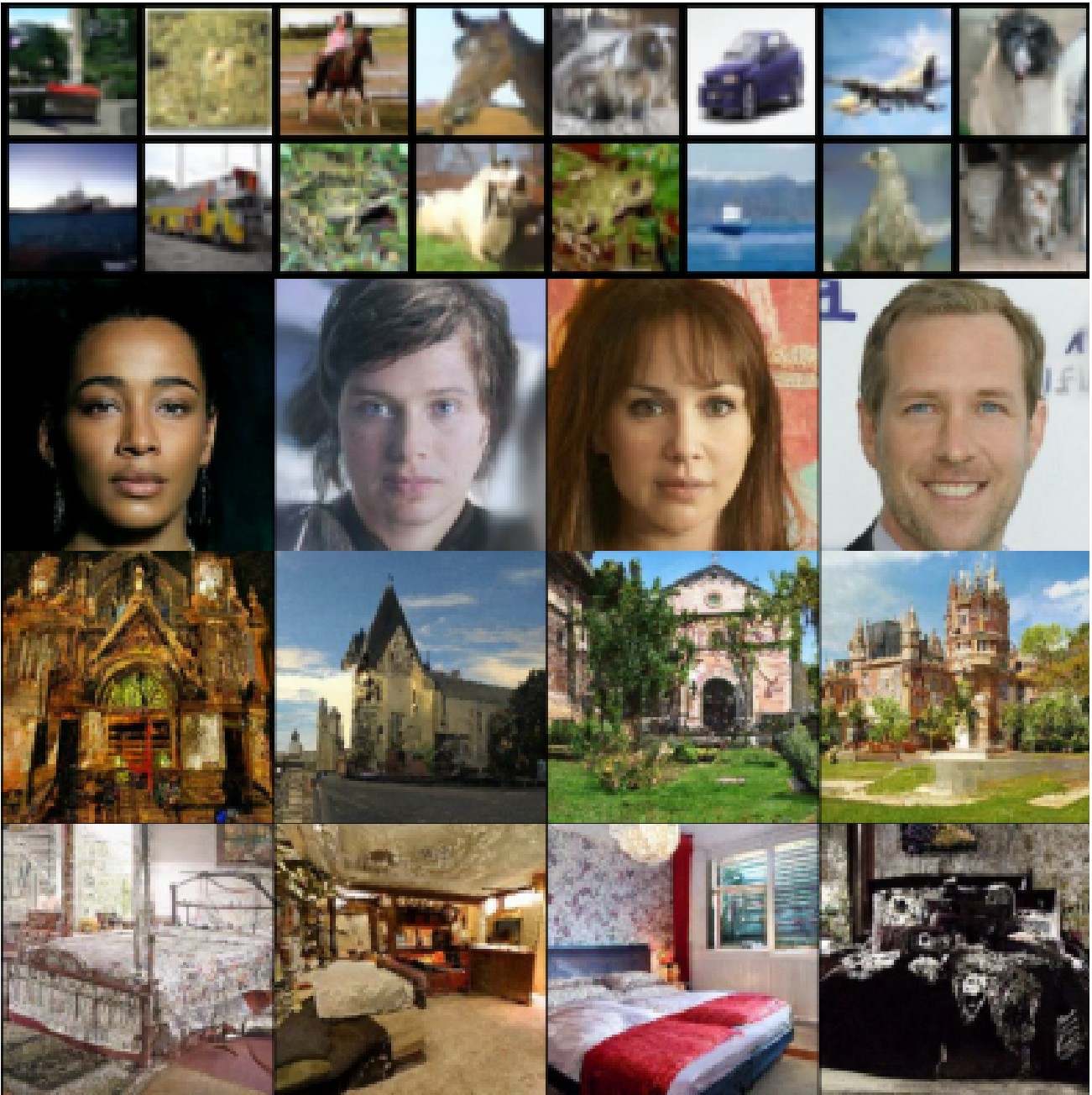

*Figure 9.* Unconditional samples generated by proposed method on 32×32 CIFAR10 (top two rows), 64×64 CelebA (upper middle), 64×64 LSUN Church (lower middle), and 64×64 LSUN Bedroom (bottom row).

**Lemma C.1.** *Let*

$$B = I_d + \sigma^2 U, \qquad \tilde{X} = B^{-1/2} X, \qquad \tilde{V} = \sigma B^{-1/2} V,$$

*where $B$ is assumed to be positive definite. Let*

$$\tilde{V}^T \tilde{V} = \Gamma \Lambda \Gamma^T$$

*be an eigen-decomposition, where $\Gamma \in \mathbb{R}^{r \times r}$ is orthogonal and $\Lambda$ is diagonal with nonnegative entries. Then, for any $X \in \mathbb{R}^d$,*

$$\left| I_d + \sigma^2 U + \sigma^2 V V^T \right| = |B| \cdot \left| I_r + \tilde{V}^T \tilde{V} \right|, \tag{65}$$

$$X^T \left( I_d + \sigma^2 U + \sigma^2 V V^T \right)^{-1} X = \tilde{X}^T \tilde{X} - (\tilde{V}^T \tilde{X})^T \left( I_r + \tilde{V}^T \tilde{V} \right)^{-1} (\tilde{V}^T \tilde{X}). \tag{66}$$

*Moreover, define*

$$L = B^{1/2} \left[ I_d + \tilde{V} \Gamma \left\{ (I_r + \Lambda)^{1/2} - I_r \right\} \Lambda^\dagger \Gamma^T \tilde{V}^T \right],$$

*where $\Lambda^\dagger$ denotes the Moore–Penrose inverse of $\Lambda$. Then $L$ is a valid square-root factor satisfying*

$$LL^T = I_d + \sigma^2 U + \sigma^2 V V^T.$$

*Equivalently, for any $X \in \mathbb{R}^d$,*

$$LX = B^{1/2} \left[ X + \tilde{V} \Gamma \left\{ (I_r + \Lambda)^{1/2} - I_r \right\} \Lambda^\dagger \Gamma^T \tilde{V}^T X \right]. \tag{67}$$

*Proof.* Equation (65) can be directly obtained by the matrix determinant lemma. For (66), denote $B = I_d + \sigma^2 U$. Applying the Sherman-Morrison-Woodbury formula yields:

$$X^T \left( I_d + \sigma^2 U + \sigma^2 V V^T \right)^{-1} X = X^T B^{-1} X - X^T \sigma^2 B^{-1} V (I_r + \sigma^2 V^T B^{-1} V)^{-1} V^T B^{-1} X,$$

followed by (66) via defining $\tilde{X} = (I_d + \sigma^2 U)^{-1/2} X$ and $\tilde{V} = \sigma (I_d + \sigma^2 U)^{-1/2} V$.

For (67), let

$$B = I_d + \sigma^2 U, \qquad A = I_d + \tilde{V} \tilde{V}^T.$$

Then

$$B + \sigma^2 V V^T = B^{1/2} A B^{1/2}.$$

Therefore, if $R = A^{1/2}$ and $L = B^{1/2} R$, then

$$LL^T = B^{1/2} R R^T B^{1/2} = B^{1/2} A B^{1/2} = B + \sigma^2 V V^T.$$

Thus, $L$ is a valid square-root factor. It remains to express $RX$ efficiently. Consider the thin singular value decomposition

$$\tilde{V} = \Upsilon \Lambda^{1/2} \Gamma^T,$$

where $\Upsilon^T \Upsilon = I_r$, $\Gamma^T \Gamma = I_r$, and $\Lambda$ is diagonal and nonnegative. Then

$$\tilde{V} \tilde{V}^T = \Upsilon \Lambda \Upsilon^T,$$

and hence

$$A = I_d + \tilde{V} \tilde{V}^T = \Upsilon (I_r + \Lambda) \Upsilon^T + (I_d - \Upsilon \Upsilon^T).$$

Consequently,

$$R = A^{1/2} = I_d + \Upsilon \left\{ (I_r + \Lambda)^{1/2} - I_r \right\} \Upsilon^T.$$

Since

$$\Upsilon = \tilde{V} \Gamma \Lambda^{\dagger/2},$$

we have, for any $X \in \mathbb{R}^d$,

$$RX = X + \tilde{V} \Gamma \Lambda^{\dagger/2} \left\{ (I_r + \Lambda)^{1/2} - I_r \right\} \Lambda^{\dagger/2} \Gamma^T \tilde{V}^T X$$

$$= X + \tilde{V} \Gamma \left\{ (I_r + \Lambda)^{1/2} - I_r \right\} \Lambda^\dagger \Gamma^T \tilde{V}^T X,$$

where the second equality uses that $\Lambda$ is diagonal. Multiplying by $B^{1/2}$ gives (67). $\square$

# D. Discussion on Time-Sampling Strategies

Regarding the time sampling strategy, our objective in (14) is derived from the path integral of the log-likelihood (Proposition 3.1), which implies a uniform integration over time. Consequently, sampling uniformly from the simplex yields an unbiased Monte Carlo estimator. While prior works often employ hand-crafted, non-uniform sampling schemes to emphasize difficult noise levels (Song et al., 2021c; Karras et al., 2022), incorporating such schedules into LM would require importance weighting to maintain unbiasedness. Exploring importance sampling or non-uniform weighting within the LM framework to reduce gradient variance remains an interesting direction for future work.

**Relation to the fixed-grid analysis.** The theoretical analysis in Section 5 is stated on the standard unit grid $t_k = k$ and $N = T$, following common practice in non-asymptotic analyses of diffusion samplers. The randomized grid used in Algorithm A is an optimization device for estimating the time-averaged LM objective. Conditional on any realized grid $\tau$, the objective is a deterministic-grid LM objective with non-uniform step sizes $\Delta_k = t_k - t_{k-1}$. Hence the same transition-level arguments apply after replacing unit-step quantities by their $\Delta_k$ analogues. The additional randomness from sampling $\tau$ contributes to stochastic-gradient variance, rather than changing the target population objective.

