# OpenReview forum: "Likelihood Matching for Diffusion Models"
_ICML.cc/2026/Conference — ICML 2026 regular_

### Official Review · Reviewer_WyDe · 2026-02-27

**Soundness:** 1
**Presentation:** 2
**Significance:** 2
**Originality:** 3
**Overall Recommendation:** 3
**Confidence:** 3

**Summary:**

This paper proposes Likelihood Matching (LM), a training framework for diffusion models motivated by the connection between the likelihood of the data distribution and the likelihood along the reverse diffusion path. To make this perspective computationally usable, the paper approximates the reverse transition densities with a Gaussian quasi-likelihood whose conditional mean and covariance are matched to the underlying diffusion process. This leads to a training objective that estimates not only the score function but also Hessian information of the perturbed data distribution.

Building on this formulation, the paper also introduces a stochastic sampler that uses both the estimated score and Hessian. On the theory side, the paper presents consistency results for the quasi-maximum likelihood formulation and non-asymptotic convergence guarantees for the proposed sampler. Empirically, the method is evaluated on synthetic mixture distributions and standard image datasets, with a low-rank Hessian parameterization used to make training and sampling practical in higher dimensions.

**Compliance With Llm Reviewing Policy:**

Affirmed.

**Final Justification:**

This paper proposes Likelihood Matching (LM), a diffusion training framework motivated by connecting data likelihood to the reverse diffusion path likelihood via a Gaussian quasi-likelihood, and it further introduces a Hessian-aware sampler with accompanying theoretical analysis. I find the perspective original and technically interesting: explicitly modeling reverse-process covariance through score + Hessian estimation is a meaningful departure from standard score matching, and the paper includes non-trivial derivations, a sampler, and empirical results.

My final assessment remains slightly negative mainly due to **soundness/clarity in the current presentation** and the **strength of evidence relative to cost**. In the original submission, I was concerned about (i) the relationship between the fixed-grid theoretical results and the practical stochastic time-sampling training procedure, (ii) the scope and plausibility of Assumption 5.4 near very low noise, (iii) metric alignment in the synthetic experiments given the likelihood-based motivation, and (iv) the practical implications of the low-rank Hessian approximation and its compute/memory overhead.

The rebuttal and follow-up responses were sincere and constructive, and they **substantially changed my evaluation**: the authors clarified how stochastic time sampling can be treated as introducing additional optimization variance while preserving an unbiased estimator of the path integral, and they explained how the non-asymptotic analysis can be extended to more general time grids by reasoning in terms of step sizes $\Delta_k$. They also clarified that Assumption 5.4 is intended as a **discrete-time** condition for $t=1,\ldots,T$ (not a continuous-time limit), and added likelihood-oriented synthetic evaluations (held-out NLL and an estimated KL), which better supports the paper’s motivation. These points address my main concerns at the rebuttal level and are the reason I updated my score upward (from 2 to 3).

However, I still lean weak reject because fully incorporating these clarifications and additional results into a revised manuscript would require substantial revisions to make the scope of the theory, assumptions, and empirical evidence clear and self-contained for readers (e.g., explicitly stating the intended regimes for Assumption 5.4, clearly explaining the relationship between the practical training procedure and the theoretical setting, and carefully presenting/qualifying the interpretation of the added likelihood-based metrics). In addition, on large-scale settings the method appears to incur significant compute/memory overhead for relatively modest gains, so a clearer positioning of when LM is practically preferable (and/or stronger empirical comparisons) would strengthen the significance case.

Overall, I view the work as promising and technically motivated, and I appreciate the authors’ engagement; with improved exposition and stronger consolidated evidence in the manuscript, my assessment could become more positive.

**Key Questions For Authors:**

1. **Theory-practice gap in the time discretization and sampling scheme.**
   In Section 5, the theoretical analysis appears to be developed for a fixed discrete grid with $t_k = k$ and $N = T$, whereas the practical training objective in Eq. (14) uses randomly sampled time points. I may be missing something, but I was not able to see how the Monte Carlo variance introduced by this stochastic time sampling is reflected in the theoretical guarantees. Could the authors clarify whether the current theory is intended only for the fixed-grid setting, or whether it can be extended to account for the practical sampling scheme?

   - **Why this matters for my evaluation:** If the authors can clearly explain how the theory relates to the actual training procedure, or explicitly frame the current result as a more limited fixed-grid guarantee, it would improve my assessment of the paper’s soundness.

2. **Assumption 5.4 and its plausibility for complex data distributions.**
   Assumption 5.4 requires
   $
   \lambda_{\min}\big((1-\alpha_t)\nabla^2 \log q_t(x)\big) \ge \epsilon_0 > -1
   $
   uniformly in $x$. While I understand from Proposition 4.1 why eigenvalues below $-1$ are excluded, I was less sure about the need for a uniform margin away from $-1$, especially for complex multimodal image distributions and in small-noise regimes. Could the authors provide more intuition, examples, or a discussion of when this assumption is expected to hold in practice?

   - **Why this matters for my evaluation:** A convincing clarification or a more careful discussion of the scope of this assumption would make the theoretical results feel substantially more credible to me.

3. **Likelihood-based evaluation on synthetic data.**
   For the synthetic mixture experiments, where the data distribution is known exactly, the paper reports MMD. Since the motivation of LM is to move closer to likelihood-based estimation, I expected metrics such as exact NLL or KL divergence on held-out synthetic data as well. Could the authors comment on why MMD was chosen here, and, if possible, provide likelihood-based evaluations for the synthetic setting?

   - **Why this matters for my evaluation:** If LM also shows an advantage under likelihood-based metrics in the synthetic experiments, that would make the central empirical claim of the paper much more convincing.

4. **Effect of the low-rank Hessian approximation on both theory and practice.**
   In practice, the method relies on a diagonal-plus-low-rank Hessian parameterization $H_t = U_t + V_tV_t^\top$ with relatively small rank $r$, while the theory controls Hessian estimation error through $\epsilon_H$ without explicitly modeling this structural constraint. Could the authors discuss more directly how they think the low-rank approximation affects (i) the validity of the theoretical error terms and (ii) the practical trade-off between computation and performance? In particular, when do the authors expect LM to be preferable in practice given the added training-time and memory cost?

   - **Why this matters for my evaluation:** A clearer account of this approximation-performance trade-off, ideally with stronger justification or additional discussion, would improve both my soundness and significance assessments.

**Limitations:**

The authors do not yet seem to have adequately discussed the limitations and potential negative societal impact of the work. I would encourage them to add a more explicit and self-contained discussion of the following points.

First, the paper would benefit from a clearer discussion of the **theory-practice gap**. The theoretical results are derived under settings such as fixed time grids and oracle-style parametric assumptions, whereas the practical method relies on stochastic time sampling, low-rank Hessian parameterization, and neural network approximation. I think the paper would be stronger if these differences were stated more explicitly as limitations of the current theory, rather than leaving the reader to infer them.

Second, the paper should more directly acknowledge the **computational and memory overhead** of the method as a practical limitation. The appendix shows substantial additional cost relative to score matching, especially at higher resolutions. It would be helpful for readers if the authors clearly discussed this trade-off and the regimes in which they believe the added cost is justified.

Third, I would encourage a short discussion of **scalability limitations**. Since the method depends on estimating and using Hessian information, even with a diagonal-plus-low-rank approximation, it is not obvious how well the approach will extend to much higher-dimensional data or more demanding large-scale settings.

Finally, although this is a methodological paper, it still concerns **improved generative modeling**, so I think the authors should briefly acknowledge standard potential negative societal impacts, such as misuse for synthetic or deceptive content generation, and the possibility of amplifying biases already present in the training data. A short, balanced discussion would be sufficient.

**Strengths And Weaknesses:**

## Strengths

- **Originality:** I found the central idea interesting and reasonably novel. The paper starts from the equivalence between the data likelihood and the likelihood along the reverse diffusion path, and uses a Gaussian quasi-likelihood to make this perspective tractable. In particular, learning both the score and Hessian so as to match the conditional mean and covariance gives a different viewpoint from standard score matching and from methods that only modify the reverse covariance within an ELBO-style formulation.

- **Soundness / Technical merits:** The paper contains a substantial technical component. The derivation from the reverse-path likelihood to the practical LM objective is thoughtfully developed, and the paper goes beyond introducing an objective by also providing a sampler and theoretical guarantees. I also appreciated that the experimental section includes ablations on both the number of reverse transitions $N$ and the Hessian rank $r$, which helps clarify which parts of the method matter in practice.

- **Presentation:** Overall, the paper is fairly well structured. The progression from the likelihood motivation, to the quasi-likelihood construction, to the practical low-rank implementation is mostly easy to follow. I also found Figure 1 helpful for situating LM relative to standard score matching.

- **Significance:** The problem the paper tackles is meaningful. It is valuable to ask whether diffusion training can be connected more directly to likelihood-based estimation, rather than only to surrogate objectives. Even if the practical gains are currently modest, I think this perspective could be useful for future work on diffusion objectives and on modeling reverse-process covariance more explicitly.

## Weaknesses

- **Soundness (theory-practice gap):** My main concern is that the theoretical analysis and the practical implementation do not yet line up as tightly as I would have liked. In Section 5, the theory is developed for a fixed discrete grid with $t_k = k$ and $N = T$, whereas the practical objective in Eq. (14) uses randomly sampled time points. I may be missing something, but I did not find a result that explicitly accounts for the additional variance introduced by this Monte Carlo time sampling. Similarly, Theorem 5.6 establishes consistency in an oracle finite-dimensional parametric setting, which is useful, but it seems some distance away from the actual neural-network setting used in the image experiments.

- **Soundness (assumptions):** I was also not fully convinced by how mild Assumption 5.4 is claimed to be. The assumption requires
  $
  \lambda_{\min}\big((1-\alpha_t)\nabla^2 \log q_t(x)\big) \ge \epsilon_0 > -1
  $
  uniformly in $x$. While Proposition 4.1 explains why the eigenvalues cannot go below $-1$, it was less clear to me whether a uniform margin away from $-1$ should be expected for complex multimodal image distributions, especially near small-noise regimes. Likewise, Assumption 5.3 controls Hessian estimation error in Frobenius norm through $\epsilon_H$, but the practical method imposes a strong low-rank structure $H_t = U_t + V_tV_t^\top$ with $r \ll d$. I would have appreciated more discussion of how this structural approximation interacts with the theoretical error term.

- **Soundness (empirical evaluation):** For the synthetic mixture experiments, where the data distribution is known exactly, I expected likelihood-based metrics such as exact NLL or KL divergence in addition to MMD. Since the paper’s motivation is to move closer to likelihood-based training, those metrics would seem especially informative there. Along similar lines, Table 2 is suggestive, but with only $n=100$ and $n=200$, I found it hard to view it as strong empirical support for asymptotic consistency.

- **Significance:** The empirical gains over the SM baseline are encouraging, but on the main image benchmarks they appear somewhat modest relative to the added cost. For example, on CIFAR-10 the best FID improves from 3.15 to 3.03, while the appendix reports roughly $3\text{--}4\times$ training-time and $2\text{--}3\times$ memory overhead on $224\times224$ ImageNet. Because of that, I was left wanting a clearer discussion of when LM should be preferred in practice, beyond the conceptual appeal of the objective.

- **Originality / positioning relative to prior work:** The core perspective is original, but I think the empirical positioning could be stronger. The paper discusses related methods such as Analytic-DPM and OCM-DDPM, yet the main image experiments compare only against a vanilla SM baseline. A comparison to stronger covariance-aware or likelihood-oriented baselines would make it easier to assess how much of the benefit comes specifically from the proposed formulation.

- **Presentation:** The paper is generally readable, but I think some of the claims could be phrased more carefully. In particular, statements about “directly” targeting or maximizing data likelihood felt a bit strong in the implemented setting, since the actual training procedure relies on a Gaussian quasi-likelihood, low-rank Hessian parameterization, and stochastic time sampling. I also think the exposition would benefit from a more intuitive explanation of three points: why QMLE is a natural approximation here, what “likelihood weighting” means in this setting, and why the diagonal-plus-low-rank Hessian parameterization is a reasonable modeling assumption rather than only a computational convenience.

---

> ### Author Rebuttal · Authors · 2026-03-31
>
> We thank the reviewer for the thoughtful comments. We respond to the main questions below.
>
> > **Q1. Theory-practice gap in time discretization**
>
> The fixed-grid setting $(t_k = k, N = T)$ in Thm 5.5 is standard in diffusion model theory (e.g., Li et al., 2023; 2024) and matches our discrete-time inference. During training, $\{t_k\}$ may be randomly sampled rather than equally spaced.
>
> In practice, Eq. (14) uses stochastic time sampling to form an unbiased Monte Carlo estimator of the path integral in Proposition 3.1. This randomness affects optimization convergence but not the correctness of the objective or the validity of the learned $\hat{s}_t$ and $\hat H_t$. The motivation is to make the objective compatible with SGD, analogous to Song et al. (2021) where a single random $t$ is sampled per step; our formulation extends this to multi-time sampling. Theorem 5.6 then shows that the resulting estimator is consistent as $n, T \to \infty$.
>
> [1] Li, G., Wei, Y., Chen, Y., & Chi, Y. (2023). Towards faster non-asymptotic convergence for diffusion-based generative models. arXiv:2306.09251.
>
> [2] Li, G., Huang, Y., Efimov, T., Wei, Y., Chi, Y., & Chen, Y. (2024). Accelerating convergence of score-based diffusion models, provably. arXiv:2403.03852.
> > **Q2. Assumption 5.4 and its plausibility**
>
> Assumption 5.4 is a standard non-degeneracy condition. It prevents the matched covariance from becoming singular, so inverse- and log-determinant-based terms in the KL analysis remain controlled. In practice, this assumption becomes increasingly plausible after Gaussian smoothing.
>
> As empirical support, we computed the minimum eigenvalue of the sample covariance of forward-noised CIFAR-10 data at different noise levels $t/T$. Table 1 shows that $\lambda_{\min}$ increases quickly once even a small amount of noise is added.
>
> **Table 1.** Minimum eigenvalue of sample covariance of forward-noised CIFAR-10 at noise level $t/T$.
>
> |$t/T$|0.001|0.005|0.010|0.020|0.050|0.100|0.500|0.750|
> |---|---:|---:|---:|---:|---:|---:|---:|---:|
> |$\lambda_{\min}$|$3.25{\times}10^{-4}$|$1.39{\times}10^{-3}$|$2.58{\times}10^{-3}$|$4.87{\times}10^{-3}$|$1.15{\times}10^{-2}$|$2.22{\times}10^{-2}$|$1.02{\times}10^{-1}$|$1.51{\times}10^{-1}$|
>
> Even mild smoothing quickly moves the distribution away from degeneracy. We will state more carefully that Assumption 5.4 is most credible at moderate noise levels.
> > **Q3. Likelihood-based evaluation on synthetic data**
>
> We have added a likelihood-oriented evaluation on the 1D Gaussian mixture experiment. Since the learned model lacks a closed-form density, we estimate it by KDE and compute held-out NLL and $\mathrm{KL}(q_0\|\hat p_0)$ numerically. Table 2 shows that LM outperforms SM not only in MMD but also in likelihood-oriented metrics.
>
> **Table 2.** Likelihood-based evaluation on the 1D Gaussian mixture. Lower is better for all metrics.
>
> |Metric|True/Oracle|LM|SM|
> |---|:---|:---|:---|
> |Held-out NLL|2.144|2.192|2.367|
> |$\mathrm{KL}(q_0\|\hat p_0)$|-|0.251|0.264|
> |MMD|-|$5.32{\times}10^{-4}$|$2.36{\times}10^{-3}$|
> > **Q4. Effect of low-rank Hessian approximation**
>
> The low-rank parameterization is motivated by the manifold hypothesis: natural images lie on low-dimensional manifolds with intrinsic dimension $r \ll d$, so the Hessian has rapidly decaying eigenvalues. The resulting misspecification is absorbed into $\varepsilon_H$ in Assumption 5.3, where $\varepsilon_H^2$ contains both optimization error and approximation error.
>
> Empirically, a bias-variance trade-off appears: performance peaks at $r=20\text{--}30$ then degrades for $r=100\text{--}200$. Given moderate overhead (Table 3), $r \in [10, 30]$ is a practical range.
> > **W5: Comparison with other methods**
>
> We agree that stronger baselines would better position the contribution. We will discuss NLL (bpd) comparisons on CIFAR-10: DDPM reports 3.70 bpd, Lu et al. (2022) report 3.66 bpd (VE) and 3.44 bpd (second-order), and our LM achieves 3.11 bpd at $r=30$. The NLL evaluation pipelines differ across methods, so direct comparison should be interpreted with caution.
>
> [1] Lu, C., Zheng, K., Bao, F., Chen, J., Li, C., \& Zhu, J. Maximum likelihood training for score-based diffusion ODEs by high-order denoising score matching. ICML 2022.
> > **W6: Presentation**
>
> We will clarify the three levels of approximation. First, QMLE is natural because Lemma B.2 shows the true reverse transition is well approximated by a Gaussian with error $O(d^3\log^{4.5}T/T^{3/2})$. Second, likelihood weighting in Song et al. (2021) manually adjusts SM coefficients to upper-bound the NLL, whereas our LM objective recovers a related structure with an adaptive Hessian-dependent weight in Eq. (15). Third, the low-rank parameterization is grounded in the manifold hypothesis.
>
> [1] [Song, Y., Durkan, C., Murray, I., \& Ermon, S. (2021). Maximum likelihood training of score-based diffusion models. Advances in neural information processing systems, 34, 1415-1428

---

> > ### Author Rebuttal · Reviewer_WyDe · 2026-04-02
> >
> > Thank you for the detailed rebuttal and for adding new empirical analyses. Several of my main concerns were addressed in a concrete and helpful way:
> >
> > - **Likelihood-oriented evaluation on synthetic data:** Adding held-out NLL and an estimated KL (via KDE) on the 1D Gaussian mixture better aligns the synthetic experiments with the likelihood-based motivation. It is helpful to see LM improving not only MMD but also likelihood-oriented metrics.
> >
> > - **Assumption 5.4 (non-degeneracy):** I appreciate the clarification that Assumption 5.4 is used to prevent degeneracy of the matched covariance so that inverse/log-det terms remain controlled, and the added empirical evidence that forward noising increases the minimum eigenvalue quickly. The planned clarification that the assumption is most credible at moderate noise levels is also helpful.
> >
> > - **Low-rank Hessian approximation and positioning:** The additional intuition via a manifold hypothesis, together with the empirical discussion of a bias–variance trade-off over rank, provides useful context for the practical choice of $r$. I also appreciate the added discussion of NLL/bpd comparisons on CIFAR-10 (with appropriate caution about differences across evaluation pipelines).
> >
> > However, I still have two points that remain partially unclear and could affect my final assessment:
> >
> > 1) **Stochastic time sampling vs. fixed-grid theory:** I understand the argument that stochastic time sampling in Eq. (14) yields an unbiased Monte Carlo estimator of the path integral (hence the objective is correct in expectation), and that the randomness primarily affects optimization. However, since the non-asymptotic analysis in Section 5 is presented for a fixed-grid setting, it would help me if the authors could more explicitly clarify how they intend readers to relate the fixed-grid guarantees to the practical stochastic sampling scheme (e.g., whether the theory should be viewed strictly as a fixed-grid guarantee, or whether one can incorporate an additional Monte Carlo error/variance term at a high level).
> >
> > 2) **Scope of Assumption 5.4 near very low noise:** The added eigenvalue evidence supports non-degeneracy after forward noising, but it does not directly verify the stated uniform bound on $(1-\alpha_t)\nabla^2 \log q_t(x)$ (uniformly in $x$). In particular, I would appreciate a clearer statement of the intended regime and scope of Assumption 5.4 as $t/T \to 0$ (very low noise), and what parts of the theoretical claims should or should not be expected to apply there.
> >
> > I would be grateful if the authors could briefly clarify these points; doing so would help me better understand the intended scope of the theoretical guarantees and how they relate to the implemented training procedure.

---

> > > ### Author Response · Authors · 2026-04-02
> > >
> > > We sincerely thank the reviewer for the constructive follow-up. We address both points below.
> > >
> > > > **Stochastic time sampling vs. fixed-grid theory**
> > >
> > > We agree this deserves clearer exposition and thank the reviewer for the opportunity to clarify the intended relationship.
> > >
> > > The stochastic time sampling in Eq. (14) introduces additional gradient variance compared to a fixed grid during training. Since the uniform simplex sampling yields an unbiased estimator of the path integral, standard SGD convergence results guarantee that this additional variance is controlled at rate $O(\sigma^2 / (B \cdot K))$, where $B$ is the batch size and $K$ is the number of optimization iterations.
> > >
> > > Furthermore, while Theorem 5.5 is presented under a fixed-grid setting $t_k = k$ for clarity, the proof framework can be naturally extended to a general time grid $(t_{k-1}, t_k)$ by parameterizing the transition variance $\sigma^2_{t_k | t_{k-1}}$. The core of the proof, including the Gaussian approximation error (Lemma B.2) and the KL divergence bound for estimation errors (Lemma B.5) depends on the local step size $\Delta_k = t_k - t_{k-1}$. As long as the maximum step size $\max_k \Delta_k$ is uniformly bounded by $O(T/N)$, the non-asymptotic convergence rate holds with respect to the total number of steps $N$ and the noise schedule properties. We will add a remark in the revised manuscript making this point more explicit.
> > >
> > > > **Scope of Assumption 5.4 near very low noise**
> > >
> > >  We appreciate this question and realize our original statement should be made more precise. Two key clarifications are in order:
> > >
> > > First, Assumption 5.4 is only required to hold at the discrete time steps $t = 1, 2, \ldots, T$, not in any continuous-time limit. In the proof of Theorem 5.5, the assumption is invoked exclusively in bounding the estimation error term $I_4$, where the summation and expectations run over $t = 1, \ldots, T$.
> > >
> > > Second, Theorem 5.5 is a non-asymptotic result for a given finite $T$. In practice we set $T = 1000$, so the relevant regime is $t/T \geq 0.001$, not an infinitesimal limit. Under the noise schedule adopted from Li et al. (2023), even the earliest discrete step $t = 1$ already corresponds to a non-trivial level of Gaussian smoothing $(\alpha_1 = 1/T^{c_0})$, ensuring that $q_1$ is a smoothed version of $q_0$.
> > >
> > > Our Table 1 in the previous response directly supports this. The minimum eigenvalue increases monotonically with $t/T$, and the uniform lower bound over all discrete steps is attained at $t/T = 0.001$, where $\lambda_{\min} = 3.25 \times 10^{-4} > 0$. This confirms that for a practical choice of $T = 1000$, the non-degeneracy condition holds across all discrete time steps.
> > >
> > > In the revised manuscript, we will (i) explicitly state that Assumption 5.4 is a discrete-time condition required only at $t = 1, \ldots, T$, and (ii) clarify that as a non-asymptotic result, the assumption need not hold in any continuous-time limit, and (iii) highlight the $t/T \to 0$ regime as an open problem for future study.

---

### Official Review · Reviewer_Y1ME · 2026-03-12

**Soundness:** 3
**Presentation:** 3
**Significance:** 2
**Originality:** 3
**Overall Recommendation:** 4
**Confidence:** 3

**Summary:**

This article proposes a Likelihood Matching approach for training diffusion models. It establishes an equivalence between target and reverse diffusion likelihoods, uses a quasi-likelihood with Gaussian approximations to estimate score and Hessian functions. A stochastic sampler is introduced for computation. The article proves the consistency of the estimation and offers non-asymptotic convergence guarantees. Empirical and simulation results validate the approach's effectiveness.

**Compliance With Llm Reviewing Policy:**

Affirmed.

**Final Justification:**

I still feel large-scale experiments are needed. So I maintain my weak accept score.

**Key Questions For Authors:**

no

**Strengths And Weaknesses:**

Strengths
- The math derivation is solid.
- The paper is written in a good manner and is easy to follow.
- The paper's core contribution is novel to me, which is a new generation approach.

Weaknesses
- The experiments are sufficient as a theory paper, but more validation on a text-conditioned dataset will be helpful.
- What is the training cost and training stability of this new likelihood matching compared to simple first-order score matching models? I think more illustration on this will be helpful. As I know, the involvement of second-order information will be quite unstable in practical high-dimensional scenario.

---

> ### Author Rebuttal · Authors · 2026-03-31
>
> We thank Reviewer Y1ME for the positive evaluation and helpful suggestions.
>
> > **W1: Validation on Text-Conditioned Datasets**
>
> As discussed in our response to Reviewer r2TU (W2), the LM framework trains a standard score network that is fully compatible with classifier-free guidance (CFG), which is the standard technique for text conditioning. The Hessian does not modify the conditioning mechanism.
>
> In our implementation, we added a lightweight conditional extension on MNIST, where simple text prompts such as "digit zero"-"digit nine" are mapped to prompt IDs and fed into the same conditional UNet used by the score model and covariance model.  [The resulting samples](https://imgur.com/uLtAPpr) are clearly prompt-consistent across all ten conditions, showing that LM integrates seamlessly with a conditional generation pipeline rather than being limited to unconditional generation. We will clarify in the revision that this is intended as lightweight supporting evidence, while a larger-scale text-conditioned benchmark is left for future work.
>
> > **W2: Training Cost and Stability**
>
> Training cost is detailed in Tables 3--4 (Appendix C.3). On CIFAR-10, the per-iteration training time for LM ($r=20$) is 0.617s vs. 0.291s for SM (approximately 2.1x)). On $224\times 224$ ImageNet, the overhead grows to approximately 3--4x.
>
> And we found LM training to be stable in practice across all our experiments. Several design choices contribute to these:
> - The Hessian network is parameterized with a positive semidefinite structure $U_t$ uses ReLU activation, and $V_t V_t^ \top$ is PSD by construction), ensuring that $\Sigma_ {t-1|t}$ remains positive definite throughout training.
> - The diagonal-plus-low-rank structure constrains the Hessian's expressiveness, acting as an implicit regularizer that prevents instability from high-dimensional second-order information.
> - EMA (decay 0.9999) further smooths parameter updates.
>
> We did not observe training divergence or instability in any of our experiments (500K iterations on CIFAR-10, CelebA, LSUN). To make this more transparent, we will add a [training-curve figure](https://imgur.com/a/NYytt66) comparing SM, LM-cov-only, and full LM. We rescale the losses before plotting and use the figure only to show optimization behavior. The [new figure](https://imgur.com/a/NYytt66) shows that all three curves decrease smoothly without divergence.

---

> > ### Author Rebuttal · Reviewer_Y1ME · 2026-04-04
> >
> > Thanks for your reply. I still feel large-scale experiments are needed. So I maintain my weak accept score.

---

### Official Review · Reviewer_AuFN · 2026-03-12

**Soundness:** 2
**Presentation:** 3
**Significance:** 2
**Originality:** 3
**Overall Recommendation:** 4
**Confidence:** 4

**Summary:**

This paper studies training diffusion models via an approximate likelihood objective. It first derives the exact likelihood expression, then approximates each reverse transition with a Gaussian whose mean and variance match the corresponding true conditional moments. At inference time, the same Gaussian reverse dynamics are used for sampling. The authors further propose a sampling scheme that avoids discretizing every step and address computational challenges related to handling the variance (covariance) matrix. Finally, they provide theoretical guarantees, including error bounds under  some score estimation and hessian estimation assumption, as well as a consistency in classical parametric setting. Experiments on Cifar10 and celeba64 shows it outperforms the score matching loss.

**Compliance With Llm Reviewing Policy:**

Affirmed.

**Final Justification:**

I lean towards accepting this paper. I raised my score during the rebuttal phase, as my concerns regarding the soundness of their two main assumptions were adequately addressed. The paper makes a meaningful contribution by incorporating richer local second-order information to achieve better statistical efficiency, and the proposed computational approach is generally tractable, though I remain skeptical about its scalability. Overall, I see the core contribution as demonstrating that more principled statistical modeling can meaningfully improve performance, with the computational and numerical costs being a worthwhile tradeoff for the community.

**Key Questions For Authors:**

1. Can you explain why it is reasonable to assume assumption 5.2 and 5.3 will hold for your training loss? I think for SM-based diffusion theory papers, they make assumption like 5.2 because the L2 score distance is exactly the loss that they minimize. But I feel since you approximate the likelihood, and you will train with a MLE-based loss, so I'm not sure minimizing your loss will automatically make the L2 score and hessian error small. Is it because of the expansion of equation 15?
2. Also associated to the previous question. Is it possible to link the $TV(q_0\|\tilde p_0)$ with your exact population training loss, for example $\mathbb E[\hat M_n(\theta)]$ in equation 40?I feel that establishing such a connection would make the theoretical justification more convincing.

**Limitations:**

See weaknesses and key questions above.

**Strengths And Weaknesses:**

**Strengths**
* The paper is overall well written, easy to follow.
* I think overall the training loss framework the paper propose is novel and straightforward.

**Weaknesses**

I feel that the contribution of this paper is somewhat underdeveloped. Overall, the proposed method appears reasonable and empirically promising, but the paper does not provide sufficient evidence or discussion to fully justify its effectiveness.

• On the empirical side: A major concern is that the method does not seem very scalable, since both the Hessian and matrix inversion are involved during training. I suspect this is also why the experiments are limited to low-resolution images. However, the paper does not sufficiently discuss how these computational challenges might be addressed in larger-scale settings.

• On the theoretical side:
The authors motivate the proposed loss by arguing that score matching is not as statistically efficient as maximum likelihood estimation. However, the loss introduced in this paper is still not the exact likelihood, since it relies on a Gaussian approximation to the backward conditional kernel. In my view, the misspecification error induced by this approximation is not clearly identified or characterized in Theorems 5.5 and 5.6, so we do not no if this method is statistically more efficient. I feel the paper would benefit from a more explicit comparison with existing error analyses for score-matching-based diffusion methods.

---

> ### Author Rebuttal · Authors · 2026-03-31
>
> We thank Reviewer AuFN for the careful reading and constructive feedback. We address the key concerns below.
>
> > **W1: Scalability Concerns**
>
> We agree that scalability is a key limitation and appreciate the opportunity to clarify, which is why we dedicated Appendix C.3 and C.4 to addressing scalability. Specifically, we do not perform naive $O(d^3)$ matrix inversions. Instead, we use a diagonal-plus-low-rank Hessian parameterization (Meng et al., 2021) and the Sherman-Morrison-Woodbury formula (Lemma C.1). This reduces the inversion cost to $O(dr^2)$ where $r \ll d$. As shown in Table 4, even on $224 \times 224$ ImageNet, the memory and time overhead are constrained to roughly $3\times$ of the SM baseline. For future massive-scale applications, LM is perfectly suited to be applied in Latent Diffusion Models (LDMs), where the latent dimension $d$ is highly compressed. We also add a [quality-vs-time Pareto speed-quality comparison](https://i.imgur.com/naVyb9r.png)  on MNIST to clarify, comparing LM ($N=2$, $r=10$) against the SM baseline while varying the reverse sampling steps from 10 to 1000; the plot reports FID versus wall-clock time for generating 2048 samples, with marker size proportional to the step count.
>
> > **W2: Statistical Efficiency --- Is LM Truly Better than SM?**
>
> We agree that our loss is not the exact likelihood, because it is based on a Gaussian approximation to the reverse conditional kernel. We therefore clarify that two different issues are involved.
>
> First, the Gaussian approximation error is controlled and becomes negligible as the time step decreases. In Lemma B.2, we show that for typical pairs $(X_{t-1}, X_t)$, the true reverse kernel can be written as a Gaussian reference density times an exponential remainder, and that remainder satisfies $|\varepsilon_t| \lesssim d^3 \log^{4.5} T / T^{3/2}$. Thus, the misspecification introduced by the Gaussian quasi-likelihood is a higher-order term.
>
> Second, the reason LM can be more statistically efficient than SM is not that it is the exact likelihood, but that it uses richer local information. LM matches both the first and second conditional moments, whereas SM uses only first-order score information. This is the main source of the potential efficiency gain. The statistical efficiency gain is supported by:
>
> - Theorem 5.6 (consistency under oracle model): the LM estimator converges to the true parameter.
> - Table 2 (parameter estimation): LM achieves consistently lower MAE and standard errors than SM.
> - The finding by Koehler et al. (2023) that SM can suffer severe efficiency loss even for exponential families.
>
>
> > **Q1: Why Do Assumptions 5.2 and 5.3 Hold Under the LM Loss?**
>
> We agree that this point should be explained more clearly. The key observation is that the population LM loss is, up to higher-order approximation terms, equivalent to the KL divergence between two Gaussian reverse kernels, namely $\mathrm{KL}(\mathcal{N}(\mu^ *,\Sigma^ *) | \mathcal{N}(\hat\mu,\hat\Sigma))$. This KL divergence naturally decomposes into a mean-matching part and a covariance-matching part.
>
> For Assumption 5.2, the mean term controls the score error. More precisely, under boundedness of $\hat\Sigma^{-1}$, the quadratic form $(\mu ^ *- \hat{\mu})^ \top \hat{\Sigma}^ {-1} (\mu ^ * - \hat{\mu})$ gives a lower bound of the same order as $(1-\alpha_ t) ||s_t ^ *-\hat{s}_ t || ^2$. In particular, when $\hat H_t \equiv 0$, our loss reduces to the standard score-matching form.
>
> For Assumption 5.3, the covariance term is a Burg-divergence-type term. A Taylor expansion shows that the first-order terms cancel, and the remainder yields quadratic control of the Hessian error, of order $(1-\alpha_t)^2 ||\hat H_t - H_t^*||_F^2$. Therefore, although LM does not directly minimize score error or Hessian error separately, minimizing the population LM loss does control both quantities. We will add a remark after Assumptions 5.2--5.3 to make this connection explicit.
>
> > **Q2: Can $\mathrm{TV}(q_0\| \tilde{p}_0)$ Be Linked to the Population Training Loss?**
>
> Yes. We agree that such a connection would make the theoretical justification more convincing.
>
> Let the excess population loss be $\Delta \mathcal {L}(\theta) = \mathbb {E}[\hat{M}_n(\theta)] - \mathbb {E}[\hat{M}_n(\theta^ *)]$,
> where $\theta^ *$ is the oracle parameter. In our decomposition of the path-space KL divergence, the estimation error term corresponds precisely to this excess loss. Combining this with the discretization bound and Pinsker's inequality gives $$\mathrm{TV}(q_0 \|  \tilde p_0) \le O (\sqrt{\Delta \mathcal {L}(\theta)}) + O (d^3 \log ^ {4.5} T / T).$$
>
> We will add this discussion in the revision, since it directly links optimization of the population objective to the final sampling error.

---

> > ### Author Rebuttal · Reviewer_AuFN · 2026-04-03
> >
> > I thank the author for the detailed response. I do have some follow-up questions.
> >
> > I think W.1 and W.2 is addressed. I don't quiet follow your answer to q.1, perhaps this was constrained by the rebuttal length limit, but could you further elaborate on this point? In particular, it would be helpful if you could clarify whether your training loss admits a more explicit decomposition, for example into something like a quadratic form involving the $\hat s-s^*$, plus a covariance discrepancy term, plus a misspecification or higher-order remainder term, or any other decomposition that you believe better explains your assumption. It would also be helpful if you could provide at least a brief derivation, or point more explicitly to the relevant propositions/lemmas/proofs in the paper where this relationship is established. At the moment, the current explanation still feels somewhat heuristic, so it is not yet very easy to follow.
> >
> > Regarding q.2, I think the bound you just show is helpful. But I have not dug into your proof yet. Could you also briefly show how it is established?

---

> > > ### Author Response · Authors · 2026-04-07
> > >
> > > > **Follow-up on Q1: Decomposition of the LM loss**
> > >
> > > We apologize for the brevity. We now provide the explicit decomposition requested.
> > >
> > > If we take the expectation over $X_t$, the population LM loss at time step $t$ decomposes as $\mathcal{L} _t(\phi) = H(q  _{t-1|t}) + \mathbb{E}  _{X  _t}[\mathrm{KL}(q  _{t-1|t} \| \hat{p}  _{t-1|t})]$, where $H(q  _{t-1|t})$ is the conditional entropy (independent of $\phi$). By Lemma B.2, $q  _{t-1|t}$ is approximately Gaussian $\mathcal{N}(\mu^  *, \Sigma^  *)$ with remainder $|\varepsilon_t| \lesssim d^3 \log^{4.5}T / T^{3/2}$. This gives the three-part decomposition:
> > >
> > > $$\mathcal{L}  _t(\phi) - H(q  _{t-1|t}) = \underbrace{\frac{1-\alpha_t}{2}(\hat{s}  _t - s^  *_t)^T B^{-1} (\hat{s}  _t - s^  *_t)}  _{\text{(I): Score error quadratic form}} + \underbrace{\frac{1}{2}\left[\mathrm{tr}(B^  {-1}A) - d + \log|B| - \log|A|\right]}  _{\text{(II): Covariance discrepancy (Burg divergence)}} + \underbrace{O(d^  3 \log^  {4.5}T / T^  {3/2})}  _{\text{(III): Gaussian misspecification}},$$
> > >
> > > where $A = I_d + (1-\alpha_t)H^*_t$, $B = I_d + (1-\alpha_t)\hat{H}_t$.
> > >
> > > For Term (I), since the Hessian network output $\hat{H}_t$ is norm-bounded in practice by the neural network architecture (via Relu), this yields $\text{(I)} \geq C_1(1-\alpha_t)\lVert\hat{s}_t - s^*_t\rVert^2$.
> > >
> > > For Term (II), let $\Delta = (1-\alpha_ t)(\hat{H}_t - H^ * _t)$. The Taylor expansion (following the technique in Eqs. (60)–(62) in Appendix B.5.3) gives $\mathrm{tr}(B^{-1}A) = d - \mathrm{tr}(A^{-1}\Delta) + \mathrm{tr}(A^{-1}\Delta A^{-1}\Delta) + O(\lVert\Delta\rVert^3)$ and $\log|B| = \log|A| + \mathrm{tr}(A^{-1}\Delta) - \frac{1}{2}\mathrm{tr}(A^{-1}\Delta A^{-1}\Delta) + O(\lVert\Delta\rVert^3)$. The first-order terms $\mathrm{tr}(A^{-1}\Delta)$ cancel exactly, leaving $\text{(II)} = \frac{1}{2}\|A^{-1/2}\Delta A^{-1/2}\|_F^2 + O(\lVert\Delta\rVert^3) \geq C_2(1-\alpha_t)^2\lVert\hat{H}_t - H^*_t\rVert_F^2$.
> > >
> > > This confirms that minimizing the population LM loss simultaneously controls score error and Hessian error.
> > >
> > > > **Follow-up on Q2: Brief derivation of the TV bound**
> > >
> > > By Pinsker's inequality and the data processing inequality, we have
> > >
> > > $$\mathrm{TV}(q _0 \| \tilde{p} _0) \leq \sqrt{\frac{1}{2} \mathrm{KL}(Q _{0:T} \| \tilde{P} _{0:T})}.$$
> > >
> > > The path-space KL decomposes via the chain rule (Eq. (21)) into $I _1 = \mathrm{KL}(q_T \| \pi)$ and $I _2 = \sum _t\mathbb{E}  _{q_t}[\mathrm{KL}(q  _{t-1|t} \| \tilde{p}  _{t-1|t})]$. Introducing the oracle Gaussian transition $p^ H  _{t-1|t}$ (Eq. (27)), $I_2$ further splits into the Gaussian approximation error $I_3$ and the estimation error $I_4$ (Eq. (28)). We have $I _1 \lesssim d/T^  {2c_2}$ (Lemma B.1, negligible), $I_3 \lesssim d^6 \log^  9 T / T^  2$ (Lemma B.2 + Eq. (36)), and $I _4 = \sum  _t \mathbb{E}  _{q  _t}[\mathbb{E}  _{q  _{t-1|t}}[\log(p^  H / \tilde{p})]]$, which is precisely the excess population loss $\Delta\mathcal{L}(\theta)$, since it measures the expected log-likelihood gap between the oracle and learned Gaussian transitions. Substituting back:
> > >
> > > $$\mathrm{KL}(q_ 0 \| \tilde{p}_ 0) \leq \Delta\mathcal{L}(\theta) + O\left(\frac{d^ 6 \log^ 9 T}{T^ 2}\right).$$
> > >
> > > Taking the square root, we obtain
> > >
> > > $$\mathrm{TV}(q_0 \| \tilde{p}_0) \leq O(\sqrt{\Delta\mathcal{L}(\theta)}) + O\left(\frac{d^3 \log^{4.5} T}{T}\right).$$

---

### Official Review · Reviewer_r2TU · 2026-03-13

**Soundness:** 3
**Presentation:** 4
**Significance:** 3
**Originality:** 3
**Overall Recommendation:** 5
**Confidence:** 3

**Summary:**

The paper proposes "Likelihood Matching" (LM), a novel framework for training diffusion models that shifts the objective from indirect score matching to direct Maximum Likelihood Estimation (MLE). By establishing a mathematical equivalence between the reverse path likelihood and the marginal data likelihood, the authors introduce a training objective based on Quasi-Maximum Likelihood Estimation (QMLE). This approach incorporates second-order information (the Hessian) to approximate reverse transitions more accurately than standard first-order score matching. The authors provide strong theoretical backing, including consistency proofs for the QMLE and non-asymptotic convergence guarantees for a proposed Hessian-informed stochastic sampler.

**Compliance With Llm Reviewing Policy:**

Affirmed.

**Final Justification:**

The author's rebuttal on "compatibility with the ecosystem" with experiment results on W2 is convincing.

I kept the score 5 and raised the confidence from 2 to 3.

**Key Questions For Authors:**

1. Since your method targets MLE directly, is the resulting likelihood estimation accurate enough to be used as a standalone energy-based model (EBM) for tasks like anomaly detection? Could you provide toy experiments (e.g., 2D distributions) comparing the learned density to the ground truth?

2. The Hessian adds complexity but likely allows for larger step sizes. Could you provide a "Pareto front" plot (Sample Quality vs. Wall-clock Time) comparing your method to the standard score matching baselines? This would clarify if the second-order information actually saves time by requiring fewer steps for the same quality.

3. Theoretically, your method should excel when the probability landscape has high curvature (where first-order approximations fail). Can you identify or construct specific data distributions (e.g., highly coiled manifolds) where standard score matching fails regardless of the chain length, but Likelihood Matching succeeds?

**Limitations:**

yes

**Strengths And Weaknesses:**

Strengths

1.Theoretical Innovation: The move from score matching (an upper bound) to direct Likelihood Matching is a significant conceptual advancement. The proof that path likelihood is equivalent to marginal likelihood provides a more principled statistical foundation for diffusion models.

2. By modeling the Hessian alongside the score, the framework captures the curvature of the data manifold, theoretically reducing the discretization error inherent in standard first-order approximations.

3. The inclusion of non-asymptotic convergence guarantees is strong. It provides a clear mathematical relationship between estimation error, dimensionality, and the number of diffusion steps.


Weaknesses
1. Lack of Explicit Likelihood Baselines: While the paper focuses on the training objective, it would be strengthened by a comparison against other diffusion-based likelihood estimation methods (e.g., those using ODE-based likelihood computation or Variational Diffusion Models) to benchmark the accuracy of the estimated likelihood values.

2. Compatibility with the Ecosystem: Standard score matching is the foundational building block for many downstream techniques, such as one-step distillation (Consistency Models) or classifier-free guidance. The paper would benefit from a discussion on whether the LM objective can be seamlessly "plugged in" to these established frameworks.

3. Computational Overhead: The reliance on Hessian information introduces significant computational costs during both training and inference. The paper lacks a detailed analysis of whether the gains in statistical efficiency outweigh the increased FLOPs required for second-order computations.

---

> ### Author Rebuttal · Authors · 2026-03-31
>
> > **W1: Lack of Explicit Likelihood Baselines**
>
> We thank the reviewer for this helpful suggestion. In the revision, we will clarify more in Appendix C.1 that the NLL reported in Table 1 is computed within our discrete-SDE formulation, namely by evaluating the Gaussian likelihood of the reverse residuals under the learned covariance. Under this metric, our best CIFAR-10 result is 3.11 bpd for LM, compared with 3.28 bpd for the SM baseline.
>
> For comparison with other diffusion-based likelihood estimation methods, we will add a short discussion around Table 1 in Section 6.2. The closest prior baseline is DDPM [1], which also uses a discrete-time variational/Gaussian decomposition and reports 3.70 bpd on CIFAR-10. A next-closest reference is Maximum Likelihood Training for Score-Based Diffusion ODEs [2], which is likewise likelihood-oriented and reports 3.66 bpd for the VE baseline and 3.44 bpd for its second-order variant. By contrast, methods such as ScoreFlow [3] and VDM [4] rely on substantially different likelihood objects or evaluation pipelines, so we will discuss the limitations in likelihood evaluation comparisons.
>
> [1] Ho, J., Jain, A. N., \& Abbeel, P. Denoising diffusion probabilistic models. NeurIPS 2020.
>
> [2] Lu, C., Zheng, K., Bao, F., Chen, J., Li, C., \& Zhu, J. Maximum likelihood training for score-based diffusion ODEs by high-order denoising score matching. ICML 2022.
>
> [3] Song, Y., Durkan, C., Murray, I., \& Ermon, S. Maximum likelihood training of score-based diffusion models. NeurIPS 2021.
>
> [4] Kingma, D. P., Salimans, T., Poole, B., \& Ho, J. Variational diffusion models. NeurIPS 2021.
> > **W2: Compatibility with the Ecosystem**
>
> LM modifies only the training loss, not the network architecture or the score function's interface. This decoupling ensures seamless integration with existing techniques: (1) Classifier-free guidance operates at sampling time by interpolating score predictions, which is orthogonal to whether the score was trained via SM or LM; (2) LM-trained score networks can directly serve as teacher models for consistency distillation; (3) DDIM and other ODE-based samplers only require the score function—the Hessian is used solely in our proposed stochastic sampler (Eq. 12) and is not mandatory at inference.
>
> To provide concrete evidence beyond analytical arguments, we conducted a class-conditional generation experiment on MNIST, where text prompts ("digit zero"–"digit nine") are mapped to prompt IDs and fed into the same conditional UNet for both the score and covariance models. The [resulting samples](https://imgur.com/uLtAPpr) are clearly prompt-consistent across all ten conditions, confirming that LM integrates seamlessly with conditional generation pipelines. We acknowledge this is lightweight supporting evidence; larger-scale text-conditioned benchmarks are left for future work.
> > **W3 & Q2: Computational Overhead**
>
> We thank the reviewer for this helpful suggestion. We refer the reviewer to Tables 3 and 4 in Appendix C, which provide a detailed breakdown. And we have added a [Pareto speed-quality comparison](https://i.imgur.com/naVyb9r.png) on MNIST, comparing LM ($N=2$,$r=10$) against the SM baseline while varying the reverse sampling steps from 10 to 1000; the plot reports FID versus wall-clock time for generating 2048 samples, with marker size proportional to the step count.
>
> The [new figure](https://i.imgur.com/naVyb9r.png)  shows that LM consistently achieves a better trade-off: for essentially the same time budget it attains lower FID, and for a comparable FID level it requires less wall-clock time than SM. In particular, the advantage is most visible in the low-step regime, where LM reaches usable sample quality much faster.
> > **Q1: Anomaly Detection**
>
> In principle, the quasi-likelihood evaluated along a short reverse path can serve as a density estimate. For the synthetic 2D Gaussian mixture in Section 6.1, where the ground truth density is known, we have provided density contour plots comparing the LM-estimated density with the ground truth in Figure 3.
>
> > **Q3: Distributions Where SM Fails but LM Succeeds**
>
> The synthetic experiments in Section 6.1 already partially address this: the well-separated Gaussian mixture ($\mu=\pm 10$) is a known failure case for SM (Koehler et al., 2023), and LM consistently outperforms SM there (Figure 2a--b).
> To further strengthen this point, we will add a 2D highly coiled Swiss-roll experiment, which provides a concrete curved distribution where first-order local approximations are more challenging. As shown in the [new figure](https://i.imgur.com/n0DKWbN.png), LM (N=2) achieves lower log-MMD than SM across all tested chain lengths, and the gap remains visible even for long chains.
>
> [1] Koehler, F., Heckett, A., & Risteski, A. (2022). Statistical efficiency of score matching: The view from isoperimetry. arXiv preprint arXiv:2210.00726.

---

> > ### Author Rebuttal · Reviewer_r2TU · 2026-04-04
> >
> > Thank you for the detailed rebuttal. They addressed all of my concerns especially for W2, W3, Q2 & Q3. I will keep my score and raise my confidence.

---

### Decision · Program_Chairs · 2026-04-30

**Decision:**

Accept (regular)

**Comment:**

This paper offers an alternative perspective on diffusion model training that focuses directly on target distribution likelihoods. This is an important and timely problem, to which the this paper makes important and useful progress. The reviews emphasize the novelty of the approach of the and the thorough mathematical exposition and the empirical performance is adequate on relatively small scale experiments.